# Reverse Engineering Human Preferences with Reinforcement Learning

**Lisa Alazraki**[*]
Imperial College London

**Tan Yi-Chern**
Cohere

**Jon Ander Campos**
Cohere

**Maximilian Mozes**
Cohere

**Marek Rei**
Imperial College London

**Max Bartolo**
Cohere

## Abstract

The capabilities of Large Language Models (LLMs) are routinely evaluated by other LLMs trained to predict human preferences. This framework—known as *LLM-as-a-judge*—is highly scalable and relatively low cost. However, it is also vulnerable to malicious exploitation, as LLM responses can be tuned to overfit the preferences of the judge. Previous work shows that the answers generated by a candidate-LLM can be edited *post hoc* to maximise the score assigned to them by a judge-LLM. In this study, we adopt a different approach and use the signal provided by judge-LLMs as a reward to adversarially tune models that generate text preambles designed to boost downstream performance. We find that frozen LLMs pipelined with these models attain higher LLM-evaluation scores than existing frameworks. Crucially, unlike other frameworks which intervene directly on the model's response, our method is virtually undetectable. We also demonstrate that the effectiveness of the tuned preamble generator transfers when the candidate-LLM and the judge-LLM are replaced with models that are not used during training. These findings raise important questions about the design of more reliable LLM-as-a-judge evaluation settings. They also demonstrate that human preferences can be reverse engineered effectively, by pipelining LLMs to optimise upstream preambles via reinforcement learning—an approach that could find future applications in diverse tasks and domains beyond adversarial attacks.

## 1 Introduction

The LLM-as-a-judge framework has largely replaced human evaluation in the large-scale assessment of LLMs [5, 42, 3, 17, 35, 40], with several widely used benchmarks now relying on this approach to judge model performance across tasks [26, 20, 21, 30, 39, 40]. Judge-LLMs are trained to predict human preferences, offering a scalable and cost-effective alternative to human annotations [26, 40]. However, prior work has shown that judge-LLMs are vulnerable to adversarial attacks aimed at artificially boosting their scores [32, 34, 38]. In particular, it is possible to find text sequences that, once appended to or substituted for a response, maximise the score awarded to it by a judge [32, 34, 41]. This type of attack intervenes *post hoc* on the text being evaluated and can thus be detected via human inspection or by computing the perplexity (PPL) of the modified response [19, 32, 34].

In this study, we investigate a different, novel approach based on reward modelling w.r.t. the judge-LLM's evaluation scores, testing both its effectiveness and detectability. Specifically, we use

---

[*]Work done while at Cohere. Correspondence to lisa.alazraki20@imperial.ac.uk.

39th Conference on Neural Information Processing Systems (NeurIPS 2025).

these scores to tune an adversarial model that generates textual preambles[2]—i.e., additional sets of instructions—to be injected into a frozen candidate-LLM, causing its responses to receive higher scores from the judge. The loss function that optimises the preamble generator is adapted from Contrastive Policy Gradient [12], but depends on rewards computed solely on the generations of the candidate-LLM and without directly observing the preambles. We refer to this technique of pipelining multiple LLMs to indirectly optimise upstream preambles in an RL fashion as *Reinforcement Learning for Reverse Engineering* (RLRE).

There are several advantages to tuning an upstream preamble generator rather than directly overfitting the candidate-LLM to the judge-LLM's rewards: (1) the specialised preamble generator can be smaller in size, hence computationally cheaper to train; (2) tuning the preambles while leaving the candidate-LLM frozen retains its original capabilities and is less likely to result in noticeable stylistic changes in its output, thus potentially making the attack harder to detect; (3) the generated preambles that align the candidate-LLM to the judge are natural language instructions, which can be analysed and interpreted; (4) once trained, the preamble generator can serve as a plug-and-play component for pipelining with different candidate-LLMs (we experiment with preamble transferability across candidate-LLMs in Section 5.1); (5) in broader contexts, tuning preambles can also serve as a means of optimising models that cannot be fine-tuned directly (e.g., those accessible only via inference APIs or for which fine-tuning would be too costly).

Indeed, we find that responses produced by candidate-LLMs pipelined with the preamble generator receive substantially higher evaluation scores from judge-LLMs compared to responses attacked with other strategies, while also eluding detection methods. In contrast, existing attacks can be detected via perplexity analysis [19] or human inspection.

Finally, we perform an analysis of the optimal preambles and find high variability in their fluency across different model pipelines. While natural language preambles enhance interpretability, our findings raise important questions about whether constraining conditioning tokens—such as preambles or reasoning tokens—to the manifold of natural language may inadvertently limit model capabilities.

This paper makes the following main contributions:

1. We show that an adversarially tuned preamble generator, pipelined with a frozen LLM, is effective at deceiving judge-LLMs into assigning higher scores. To the best of our knowledge, this is the first work that optimises preambles to be injected into a frozen LLM for this purpose. In contrast, previous studies focus on finding text sequences to be appended to pre-generated responses.

2. We demonstrate that our adversarial preamble generator can be successfully pipelined with candidate-LLMs and judge-LLMs not seen during training.

3. We show that our attack does not increase PPL scores and is rarely flagged by human evaluators. Hence, it cannot be detected using existing safeguards.

4. We observe variations in optimal preamble style, fluency and naturalness across models, suggesting that conditioning LLMs on human-readable sequences only (for example in preambles or reasoning traces) may be overly restrictive from a performance perspective.

5. Our work highlights intrinsic vulnerabilities in the LLM-as-a-judge paradigm and calls into question its reliability and robustness.

6. More broadly, this work introduces RLRE, a novel approach that pipelines LLMs to optimise upstream textual preambles in a reinforcement learning setting. While here we use RLRE to reverse engineer human preferences with the aim of boosting LLM-as-a-judge evaluation, we postulate that this method could be paired with different downstream rewards to optimise preambles for a variety of applications beyond just adversarial attacks—including but not limited to meaningful tasks such as reducing toxicity or mitigating bias.

## 2 Related Work

Prior work investigating the robustness of LLM-as-a-judge has found that this approach suffers from multiple inherent biases. Existing research has sought to exploit these biases by crafting adversarial attacks aimed at maximising the scores assigned by judge-LLMs to candidate responses.

---

[2]In this context, preambles are also known as system prompts.

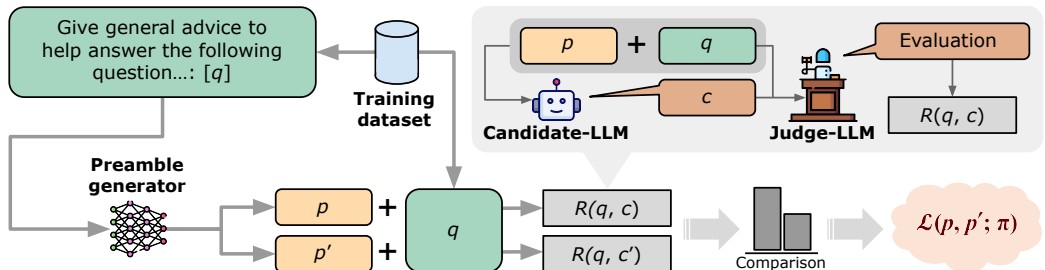

Figure 1: Reinforcement Learning for Reverse Engineering (RLRE) pipeline for training a preamble generator. Given a question $q$ from a training set, we prepend to it a general instruction and feed it to the preamble generator $\pi$. In order for $\pi$ to learn the policy, we sample two preambles per question, $p$ and $p'$. The respective rewards are obtained by appending $q$ to $p$ and $p'$, respectively, and (i) passing each as input to the candidate-LLM, which generates the responses $c$ and $c'$, and (ii) having the judge-LLM evaluate each question-response pair and extracting the respective numerical rewards from these evaluations. The loss function that optimises $\pi$ depends on the delta between the rewards $R(q,c)$ and $R(q,c')$.

**Biases.** Judge-LLMs are not unbiased evaluators. [24] and [28] observe that judge-LLMs prefer their own generations to those of other models in the large majority of cases. [36] further show that when asked to choose the best among multiple responses, GPT models favour the first candidate displayed, regardless of its quality. Additionally, [4] find that LLMs tasked with scoring other models prefer visually appealing generations regardless of content, and generations that are attributed, even falsely, to authority figures. Similarly, [38] conclude that LLM judgment is vulnerable to spurious attributions ('authority bias'). They additionally observe that judge-LLMs tend to prefer longer responses ('verbosity bias'), responses falsely identified as majority beliefs ('bandwagon-effect bias'), and responses presented as the result of a refinement process ('refinement-aware bias').

**Adversarial attacks.** [38] show that the biases of judge-LLMs can be manipulated to artificially boost their evaluation scores. They append specific text sequences to candidate responses: a false book citation to exploit authority bias, a sentence stating that most people favour that response to exploit bandwagon-effect bias, or a piece of text suggesting the response has been through a refinement process, to leverage refinement-aware bias. In each case, they find that the resulting responses are evaluated more favourably by all or some of the judge-LLMs. They also show that evaluation is affected when the length of a response is increased, without any improvement in its quality. [32] take this type of adversarial attack further, tuning a universal text sequence that, when appended to a pre-generated response, increases its evaluation score. This sequence is found by searching sequentially through the vocabulary, iteratively selecting the word that maximises the average reward from the judge on the training set. Their method is successful at inflating LLM judgement on Topical-Chat [14] and SummEval [11], and they show the attack transfers to previously unseen judge-LLMs not included in the search process. Rather than tuning a universal phrase, [34] train a sample-specific text sequence to be selected more often by a judge in pairwise comparison. Similar to [32], they append the tuned sequence to a pre-generated LLM response. Finally, [41] experiment with replacing responses with fixed instructions that invalidate the original LLM-as-a-judge prompt. Note that [32], [34] and [41] all observe that their attack can in large part be detected by measuring the perplexity of the responses. As the attack intervenes directly on the response and alters it, attacked responses tend to display higher PPL.

Unlike the above methods, the attack we propose does not modify the generated text *post hoc*. This makes its detection substantially more difficult.

## 3 Method

Given a training dataset of questions $\mathcal{D} = \{(q_j)_{1 \leq j \leq N}\}$ and a fixed instruction prompt $i$, we aim to train a preamble generator $\pi_\theta(p_j | i, q_j)$ to generate textual preambles $p$ conditioned on $i$ and $q$. We formulate the RL problem as:

$$J(\pi_\theta) = \mathbb{E}_{q \sim \mathcal{D}} \mathbb{E}_{p \sim \pi_\theta(p|i,q)} \mathbb{E}_{c \sim LLM_C(c|p,q)} [R(q,c)]$$

where $LLM_C$ is a frozen LLM—referred to as the candidate-LLM—which takes a preamble $p_j$ and the corresponding question $q_j$ and outputs a candidate response $c_j$. Note that the reward is a function of the preamble because $LLM_C$ is conditioned on it. Our reward model is a frozen LLM that outputs a verbal critique followed by a numerical score, as in the LLM-as-a-judge framework. We refer to this model as the judge-LLM. In our case, the score output by the judge is discrete on the 1–10 scale, elicited using MT-Bench [40] prompts for single (as opposed to pairwise) evaluation. We use this numerical score as the reward in our training pipeline. Figure 1 illustrates the training pipeline in detail.

In order to optimise the RL problem, we adapt Contrastive Policy Gradient (CoPG) [12]. The rationale behind this choice, as well as a comparison with other RL algorithms, is further elaborated in Appendix A. For a pair of two sampled preambles $p_j$ and $p'_j$ we introduce the following sampling loss:

$$\mathcal{L}(p_j, p'_j; \pi) = \left( R(q_j, c_j) - R(q_j, c'_j) - \beta \left( \ln \frac{\pi(p_j|i, q_j)}{\pi_{\text{ref}}(p_j|i, q_j)} - \ln \frac{\pi(p'_j|i, q_j)}{\pi_{\text{ref}}(p'_j|i, q_j)} \right) \right)^2.$$

$\pi_{\text{ref}}$ is a reference model used for regularising the RL problem, which we set to be the base LLM underlying the preamble generator. As in [12], $\beta$ is a hyperparameter regulating the importance of the KL-divergence between the sequence log-likelihoods of $\pi$ and $\pi_{\text{ref}}$ in the overall loss.

As $\pi$ is trained to generate question-specific preambles, we pipeline it with the candidate-LLM at test time to dynamically generate responses to new questions. We prompt the preamble generator with the same fixed instruction $i$ used during training, along with a question $q$, to generate the response $c$. A judge-LLM is then prompted to assign a score to $c$.

As an additional consideration, it is worth noting that [12] investigate CoPG solely in an offline manner. However, they hypothesise that the method should also scale to the online setting. To the best of our knowledge, this work is the first to successfully apply a similar method to online learning.

## 4 Experiments

### 4.1 Models and Hyperparameters

To test the generalisability of our method, we use LLMs from both the Command[3] and the Llama 3.1 [15] model series. We train and test three distinct pipelines, illustrated in Table 1. Note that all pipelines are trained with the same judge-LLM, i.e. Command R+ prompted as in [40]. We tune the Command R7B [7] preamble generators on a Google Cloud TPU v5e containing 64 chips. We train the Llama 3.1 8B Instruct preamble generator on a single Nvidia H100 GPU. Candidate- and judge-LLMs from the Command family are accessed via API. The Llama 3.1 70B Instruct candidate-LLM is deployed and queried on a local server.

Due to computational limitations, we perform all hyperparameter tuning on the `Command R7B+R7B` pipeline, and apply the same hyperparameters to the other two. As shown in Section 4.4, the `Command R7B+R7B` pipeline attains the greatest performance improvements over the baselines, and it is therefore likely that additional hyperparameter tuning would further raise the scores obtained by the `Command R7B+R` and `Llama 8B+70B` pipelines. This may be especially true for the latter, which comprises a family of models different from those used for hyperparameter tuning. It is worth noting that, since the preambles only need to influence the candidate-LLM outputs and do not necessarily need to be fluent from a human perspective, the hyperparameter tuning process results in a relatively low value of the KL-divergence coefficient ($\beta = 0.03$), which regulates the similarity between the output distributions of the preamble generator and the reference model. This low $\beta$ value allows our preambles to deviate more drastically from the reference policy during training.

We train each pipeline with early stopping according to validation performance. All hyperparameters, API model IDs, and training process details are given in Appendix B.

---

[3] https://cohere.com/command

Table 1: The training pipelines include models of different sizes and families as preamble generators and/or candidate-LLMs. Command R+ (104B parameters) is used as the judge-LLM in all pipelines.

| Pipeline identifier | Preamble generator | Candidate-LLM |
|---|---|---|
| `Command R7B+R7B` | Command R7B | Command R7B |
| `Command R7B+R` | Command R7B | Command R (35B) |
| `Llama 8B+70B` | Llama 3.1 8B Instruct | Llama 3.1 70B Instruct |

## 4.2 Datasets

We test all pipelines on MT-Bench [40], which consists of 160 open-ended questions, split among two conversational turns, grounded in the following domains: *writing*, *roleplay*, *reasoning*, *math*, *coding*, *extraction*, *STEM*, and *humanities*. We choose this benchmark as it is established and widely used in LLM assessment, and tests a balanced distribution of diverse skills on challenging multi-turn questions [40]. Crucially, MT-Bench supports independent judgement as opposed to pairwise comparison with other models [22, 21]. It is also worth noting that MT-Bench represents the setup we target in our approach, which makes it suitable to demonstrate that it is possible to reverse engineer human feedback in a controlled setting. Since MT-Bench does not comprise a training set, we fine-tune and validate the preamble generators using questions from UltraFeedback [8] (using MT-Bench prompts for single evaluation to elicit the downstream rewards). The questions in UltraFeedback are extracted from ShareGPT [6], FLAN [25], Evol-Instruct [37], UltraChat [10], FalseQA [16], and TruthfulQA [23]. Collectively, these datasets cover a wide range of topics, and the domains in MT-Bench are represented in UltraFeedback. The distribution of the different tasks within the training data is further analysed in Appendix C.2.

Additionally, we test transferability to a further benchmark, Arena-Hard [21], as discussed in Section 5.1.

## 4.3 Baselines

We compare the evaluation scores assigned to candidate-LLMs attacked with the preamble generator with those given to unattacked candidates. Additionally, we compare against existing methods that exploit vulnerabilities in the LLM-as-a-judge framework to artificially boost their evaluation scores. We describe these additional baselines below. Note that all of them modify pre-generated responses from the unattacked model.

**Verbosity bias attack.** We ask the candidate-LLM to increase the length of a pre-generated response. To this end, we use the prompt designed by [38] to lengthen the response without necessarily improving its quality.

**Bandwagon-effect bias attack.** We append to each pre-generated response a text sequence stating that a high percentage of people think that response should be awarded the highest rating. Consistent with [38], we randomly choose percentages between $60\%$ and $90\%$.

**Authority bias attack.** Using the same prompting strategy as [38], we ask the candidate-LLM to invent a plausible book source for a pre-generated response, given a citation template. The citation is then appended to the response to increase its perceived authority.

**Refinement-aware bias attack.** Given a pre-generated response, we have the candidate-LLM polish it using the refinement prompt in [38]. The judge is then presented with the original response, followed by the refinement prompt and the new, polished response.

**Universal adversarial attack.** We append to each pre-generated response the universal adversarial attack phrase from [32], learned for absolute assessment when attacking the Topical-Chat [14] overall score. Like UltraFeedback and MT-Bench, Topical-Chat encompasses a wide range of topics and features multi-turn questions similar to those in MT-Bench. Note that learning a new universal phrase on our UltraFeedback training set is computationally infeasible: for $n$ training samples, the exhaustive

Table 2: MT-Bench evaluation scores assigned by the Command R+ judge-LLM to candidate-LLMs attacked with different strategies. Each setup is run five times and the scores are averaged (showing the standard deviation in the subscript) to account for small variations due to temperature sampling.

| Candidate-LLM | | Attack type | | | | | | Preambles |
|---|---|---|---|---|---|---|---|---|
| | | No attack | Verbosity | Bandwagon | Authority | Refinement | Universal | |
| *Command R7B* | Turn 1 | $7.60_{0.07}$ | $7.33_{0.08}$ | $7.47_{0.05}$ | $7.51_{0.05}$ | $7.78_{0.05}$ | $7.58_{0.06}$ | $\mathbf{8.21}_{0.07}$ |
| | Turn 2 | $6.99_{0.08}$ | $7.29_{0.02}$ | $7.17_{0.08}$ | $7.29_{0.10}$ | $7.45_{0.08}$ | $7.25_{0.03}$ | $\mathbf{7.66}_{0.09}$ |
| | Overall | $7.29_{0.08}$ | $7.31_{0.05}$ | $7.32_{0.06}$ | $7.40_{0.07}$ | $7.61_{0.06}$ | $7.41_{0.04}$ | $\mathbf{7.93}_{0.08}$ |
| *Command R* | Turn 1 | $8.09_{0.08}$ | $7.99_{0.10}$ | $7.98_{0.06}$ | $8.17_{0.08}$ | $8.10_{0.05}$ | $8.10_{0.06}$ | $\mathbf{8.45}_{0.07}$ |
| | Turn 2 | $7.57_{0.12}$ | $7.73_{0.08}$ | $7.72_{0.14}$ | $7.65_{0.06}$ | $7.81_{0.05}$ | $7.75_{0.09}$ | $\mathbf{7.92}_{0.03}$ |
| | Overall | $7.83_{0.10}$ | $7.86_{0.09}$ | $7.85_{0.10}$ | $7.91_{0.07}$ | $7.95_{0.05}$ | $7.92_{0.07}$ | $\mathbf{8.18}_{0.05}$ |
| *Llama 3.1 70B Instruct* | Turn 1 | $8.47_{0.08}$ | $8.29_{0.06}$ | $8.39_{0.05}$ | $8.38_{0.07}$ | $8.51_{0.07}$ | $8.50_{0.05}$ | $\mathbf{8.56}_{0.08}$ |
| | Turn 2 | $7.64_{0.06}$ | $7.49_{0.08}$ | $7.65_{0.08}$ | $7.62_{0.07}$ | $7.75_{0.09}$ | $7.85_{0.07}$ | $\mathbf{7.88}_{0.08}$ |
| | Overall | $8.06_{0.07}$ | $7.89_{0.07}$ | $8.02_{0.07}$ | $8.00_{0.07}$ | $8.13_{0.08}$ | $8.17_{0.06}$ | $\mathbf{8.22}_{0.08}$ |

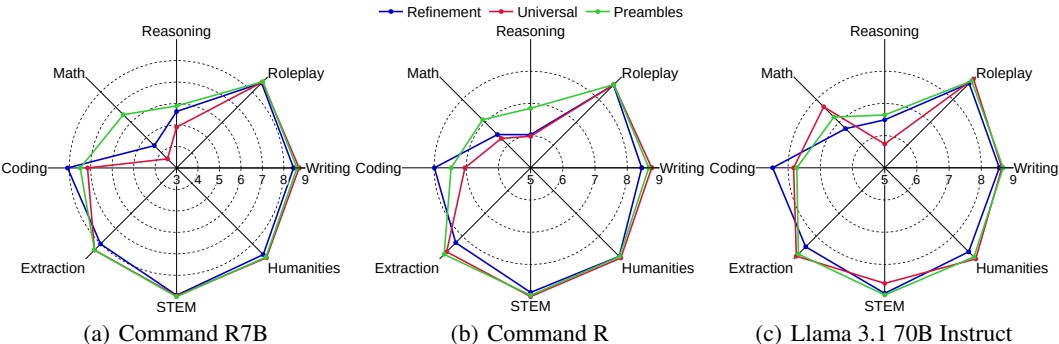

(a) Command R7B      (b) Command R      (c) Llama 3.1 70B Instruct

Figure 2: Average scores per question type obtained by candidate-LLMs using a refinement-aware bias attack, [32]'s universal adversarial attack, and the adversarial preamble generator.

search method in [32] computed over the 20k-word vocabulary makes $20000 \times 4 \times n$ calls to the judge-LLM. In our case, that would equal $4 \times 10^9$ inference calls.

## 4.4 Results

Table 2 shows the average evaluation scores obtained by three candidate-LLMs, under all baseline settings and when pipelined with our adversarial preamble generators. Models injected with adversarial preambles consistently obtain higher evaluations from the Command R+ judge at both question turns, with the `Command R7B+R7B` pipeline (i.e., the one on which we perform hyperparameter tuning) achieving the most substantial improvements. We also find that turn 1 responses benefit the most from the preambles. Note that in MT-Bench, turn 2 questions require adjusting turn 1 answers according to new constraints. This type of task is not well represented in UltraFeedback, making it OOD w.r.t. the training data (see also Appendix C.2). Nevertheless, the tuned preamble generators remain effective at raising turn 2 scores at test time.

We find that among all candidate LLMs, the strongest baseline attacks are the refinement-aware bias attack and the universal adversarial attack proposed by [32]. On average, our adversarial preambles increase Command R7B's overall score (on a 1–10 continuous scale) by 0.32 points over the refinement-aware attack and by 0.52 points over the universal attack. For Command R, the corresponding improvements are +0.23 and +0.26, respectively. Injecting adversarial preambles into Llama 3.1 70B Instruct yields smaller but consistent gains, raising its score by 0.09 relative to the refinement-aware attack and 0.05 relative to the universal attack. Furthermore, compared with their non-attacked counterparts, the preamble-based attack substantially increases the overall judge scores for Command R7B, Command R, and Llama—by 0.64, 0.35, and 0.16, respectively. We report confidence intervals in Appendix D.1.

Table 3: Candidate transferability (a) and judge transferability (b) of our preamble-based attack on MT-Bench. All scores are averaged over five runs.

(a) Candidate transferability

| Candidate-LLM | Preambles from pipeline | | |
| --- | --- | --- | --- |
| | Command R7B+R7B | Command R7B+R | Llama 8B+70B |
| *Command R7B* | $\mathbf{7.93}_{0.08}$ | $\underline{7.68}_{0.08}$ | $7.40_{0.10}$ |
| *Command R (35B)* | $\underline{8.01}_{0.09}$ | $\mathbf{8.18}_{0.05}$ | $\underline{7.97}_{0.09}$ |
| *Llama 3.1 70B Instruct* | $\underline{8.21}_{0.05}$ | $\underline{8.19}_{0.08}$ | $\mathbf{8.22}_{0.08}$ |

(b) Judge transferability

| Attack type | GPT-3.5 | GPT-4o-mini | Claude |
| --- | --- | --- | --- |
| No attack | $7.58_{0.08}$ | $6.40_{0.07}$ | $9.02_{0.06}$ |
| Verbosity | $7.36_{0.09}$ | $5.61_{0.04}$ | $8.74_{0.04}$ |
| Bandwagon | $7.47_{0.04}$ | $6.25_{0.04}$ | $8.79_{0.07}$ |
| Authority | $7.48_{0.09}$ | $5.92_{0.06}$ | $8.92_{0.12}$ |
| Refinement | $7.71_{0.11}$ | $6.39_{0.05}$ | $9.18_{0.06}$ |
| Universal | $7.33_{0.10}$ | $6.06_{0.03}$ | $8.94_{0.07}$ |
| Preambles | $\mathbf{8.07}_{0.07}$ | $\mathbf{6.71}_{0.02}$ | $\mathbf{9.44}_{0.06}$ |

In Figure 2, we illustrate the evaluation scores per question type for the two best-performing baselines (refinement-aware bias and universal adversarial attack) and the adversarial preamble generator. Overall, the adversarially tuned preambles are most effective at raising the scores of reasoning and math responses, followed by extraction, STEM and roleplay. Fine-grained results for all domains are shown in Appendix D.2.

# 5 Analysis

## 5.1 Attack Transferability

We investigate the transferability of our preamble attack across different candidate- and judge-LLMs.

**Transferability across candidate-LLMs.** At test time, we pipeline each candidate-LLM with adversarial preamble generators tuned with a *different* candidate. As shown in Table 3(a), the judge's evaluation scores remain higher than all baselines (shown in Table 2) in all cases except for Command R7B pipelined with the preamble generator from the Llama pipeline, whose scores align with [32]'s universal attack. This demonstrates strong transferability of the adversarial preamble attack across different candidate-LLMs.

**Transferability across judge-LLMs.** In real-world scenarios, the target judge-LLM may not be known in advance. It is therefore important to assess whether our method generalises to judges that were not used during training. Table 3(b) presents results where *different* judge-LLMs evaluate candidate responses across all attack settings. For this analysis, we employ GPT-3.5,[4] GPT-4o-mini [27], and Claude Haiku [2]. To maintain cost efficiency, we restrict evaluation to Command R7B. While the absolute reward scales differ across the three judges, all three consistently assign the highest scores to responses generated using our preamble generator, despite it having been trained with rewards from a different judge (Command R+). These findings suggest that the attack remains effective even when the target judge-LLM is unknown.

**Transferability across benchmarks.** We further evaluate the Command R7B pipeline on the Arena-Hard benchmark, which differs from both UltraFeedback and MT-Bench in task distribution. Moreover, its reward metric—the Arena Score Rate (ASR)—differs in both assignment strategy and numerical range, as described in [21]. We conduct evaluations using both the Command R+ judge (used during training) and an unseen judge, GPT-4, which serves as the standard evaluator for Arena-Hard at the time of writing. Applying the tuned preambles yields an average score increase of +3.5 with Command R+ and +1.2 with GPT-4 (note that Arena-Hard scores range from 0–100; see Appendix D.3 for full results). These results demonstrate that RLRE generalises effectively to new benchmarks without additional task-specific training.

---

[4] https://platform.openai.com/docs/models/gpt-3.5-turbo

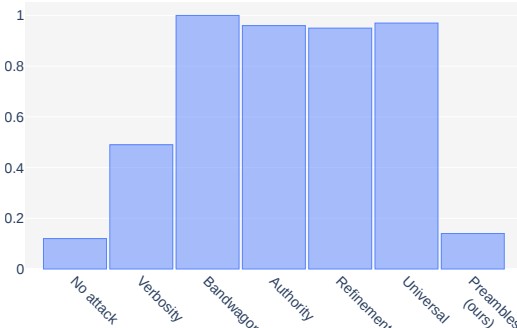

Figure 3: Proportion of LLM responses that have been labelled as 'attacked' by human evaluators. Responses generated using adversarial preambles are identified as attacked nearly as rarely as those produced by a non-attacked model.

Table 4: False negative rate (FNR) of PPL-W for each attack type. Verbosity bias, bandwagon bias and our preamble-based attack are rarely detected with this method (FNR $\geq 0.90$). The universal adversarial attack [32] is almost always detected (FNR = 0.04).

| Attack type | PPL-W (FNR) |
|---|---|
| Verbosity | 0.91 |
| Bandwagon | 0.93 |
| Authority | 0.88 |
| Refinement | 0.66 |
| Universal | 0.04 |
| Preambles | 0.90 |

## 5.2 Attack Detectability

We examine the detectability of all attacks via perplexity analysis [19]. This is the established detection method for adversarial attacks on LLM output, used in prior and contemporary work [32, 41]. Notably, this method can be used on models served via API, as is the case for our frozen candidate-LLMs, without requiring access to internal mechanisms such as attention maps [18, 29] or the entire probability distributions over the vocabulary at each decoding step [9]. As an additional test, we also measure the detectability of the attacks via human evaluation of the candidate responses.

**Perplexity analysis.** We use a perplexity filter over a 10-token sliding window as in [19], to identify whether the candidate-LLM has been attacked. A response is labelled as attacked if the PPL of any subsequence in the sliding window is greater than a threshold $t$. We set $t$ to ensure a false positive rate within 0.05 on the test set responses produced by the non-attacked model. Table 4 illustrates the results of this detection method—referred to as PPL-W—for the Command R7B candidate.

**Human evaluation.** To assess the detectability of each attack by visual inspection, we recruit 16 annotators with domain expertise and knowledge of the possible attacks. We sample a balanced subset of 400 test responses generated by Command R7B under each attack setup (including the non-attacked model), randomly split it into 16 equal-sized portions, and have each annotator assign a binary label ('attacked' or 'not attacked') to each generation. Figure 3 shows the proportion of responses labelled as 'attacked' for each attack type. Attacks that insert formulaic sequences into the response (bandwagon bias, authority bias, refinement-aware bias and universal attack) are detected in all or most cases (100%, 96%, 95% and 97%, respectively), while the verbosity bias attack is detected approximately half of the time (49%). Notably, responses produced by the preamble-based attack are labelled as 'attacked' nearly as rarely (14%) as those generated by the non-attacked model (12%). We also measure the rate at which each annotator is able to correctly detect each type of attack (both in terms of accuracy and F1). For both scores, we compute the median absolute deviation (MAD) across all annotators for each attack type, and find it to be zero in all cases, indicating that the annotators are able (or unable) to detect each attack at very similar rates. This supports the hypothesis that different attacks are inherently (un)detectable by visual inspection, as we do not find high variance in detection rates between different annotators.

## 5.3 Ablations

In our framework, we input into the preamble generator a fixed instruction concatenated with a question from the dataset, resulting in question-specific preambles. To evaluate the degree to which this prompting strategy contributes to the effectiveness of our attack, we run an ablation study where (i) we remove the question from the input and feed the preamble generator only a generic instruction, and (ii) we discard the instruction altogether and only input special tokens to signal the start of a generation turn (both prompts are shown in Appendix E). In the latter case, the model is

Table 5: Ablated MT-Bench overall scores obtained by (i) removing the question and feeding a generic instruction to the preamble generator, and (ii) removing the instruction and only feeding special tokens to signal the start of the turn. Scores are assigned by a Command R+ judge and are averaged over five runs.

| Pipeline | Ablated setting | |
|---|---|---|
| | No question | No instruction |
| Command R7B+R7B | $7.76_{0.07}$ | $7.70_{0.07}$ |
| Command R7B+R | $8.14_{0.08}$ | $8.13_{0.13}$ |
| Llama 8B+70B | $8.19_{0.11}$ | $8.18_{0.12}$ |

virtually unconstrained and may generate any text string, and thus the training signal is even more consequential in determining the content of the preambles.

Table 5 illustrates the overall scores assigned by the Command R+ judge under all ablated settings. Although both ablation strategies result in slightly lower performance than the corresponding non-ablated pipelines, all the scores remain above all baselines (refer to Table 2 for the baseline results). This demonstrates the robustness of the attack under different prompting strategies.

## 5.4 Analysis of Generated Texts

**Do successful preambles exhibit common patterns?**  Upon analysing the successful preambles produced by each tuned generator, we find consistency among preambles generated by the same model, but high variability across different models. While preambles generated by the Command models (both conditioned on the data point and conditioned on a generic instruction) share a similar structure—providing a blueprint for how to design an answer—preambles that are only conditioned on the start-of-turn token, as well as all the Llama preambles, deviate substantially from this pattern. The latter tend to reiterate the same phrases multiple times, and often do not appear fluent to a human reader. When they are not conditioned on an instruction, they even devolve into apparently meaningless sequences of characters. Nevertheless, all of these preambles are successful at raising LLM evaluation scores, as evidenced in Section 5.3. Appendix G shows representative preambles for each pipeline.

It is worth noting that, as further discussed in Appendix B, we assign relatively low weight to the log-likelihoods of the adversarial preambles within the loss function, thus prioritising reward over fluency during training. Remarkably, the training still converges toward preambles that elicit high-reward candidate responses, regardless of their fluency. This suggests that constraining preambles—or other conditioning tokens such as reasoning tokens—to the manifold of natural language may not always be optimal.

**Are attacked responses more accurate?**  Figure 2 shows that the preamble-based attack is particularly effective at raising the scores assigned to math and reasoning responses. Since these responses are evaluated by the judge-LLM against a ground truth [40], this raises the question of whether the attack has improved response accuracy. We thus evaluate via normalised exact match all math and reasoning responses from all models, both in the non-attacked setup and the preamble-attacked version. We find that the average accuracy rates are very similar, with a slight advantage for the non-attacked models (45.9%) over the preamble attack (44.2%). Hence, the accuracy of the final answer does not improve due to the preambles. However, as shown by the representative examples in Appendix H, the attacked responses have more structured reasoning chains, usually arranged into distinct paragraphs labelled with clear, explanatory headers. We thus postulate that the improved layout may be solely responsible for the higher scores assigned by the judge-LLM, regardless of correctness.

**Are attacked responses *better* overall?**  Aside from math and reasoning, the other MT-Bench domains are either not reliably verifiable with automated methods (extraction, STEM, humanities, coding) or lack objective verification entirely (writing, roleplay). To understand whether responses produced via our preamble pipeline have improved over those generated by the non-attacked model,

Table 6: Human evaluation of generated responses for all task domains.

| Attack type | Number of assigned labels | | | Avg. human rating |
| --- | --- | --- | --- | --- |
| | Poor | Fair | Good | |
| No attack | 72 | 50 | 118 | 6.22 |
| Preambles | 73 | 54 | 113 | 6.24 |

we have them evaluated by expert human annotators, blind to the attack and instructed to assess correctness and quality by assigning to each response one of three labels: *'Poor'*, *'Fair'* or *'Good'*, along with a discrete 1-10 rating matching the MT-Bench judge-LLM scale. We do this for all responses generated by Command R7B, across all domains. Table 6 shows that the label distributions are fairly similar for both the non-attacked and the attacked model, with the non-attacked model receiving slightly more positive labels (*'Good'*), and the attacked one receiving slightly more mid-range labels (*'Fair'*). We observe relatively strong inter-annotator agreement (aggregated Spearman's $\rho = 0.78$, combined $p < 0.001$, Kendall's $W = 0.72$, $p < 0.001$). Moreover, the difference between the human ratings of the non-attacked and the attacked model is negligible (0.02), whereas the difference between the judge-LLM's ratings of the same two setups is substantially greater and favours our attack.

## 6 Conclusion

We have shown that human preferences can be successfully reverse engineered to tune adversarial text preambles, and that injecting these preambles into a candidate-LLM constitutes a powerful attack on the LLM-as-a-judge framework. This attack not only outperforms previous methods at inflating the scores assigned by a judge-LLM to candidate responses, but also remains virtually undetectable using existing safeguards. In contrast, current strategies that intervene *post hoc* on the responses display high PPL and can be easily detected by visual inspection. Additionally, we have found that the attack transfers to candidate-LLMs not seen during training, enabling our adversarial preamble generator to serve as a plug-and-play component for artificially boosting the scores of multiple LLMs after a single training process. Finally, we have shown that preambles tuned with one judge-LLM can effectively attack different judges from different model families, and are effective at inflating scores on multiple benchmarks. These findings raise important questions regarding the reliability of LLM-as-a-judge evaluation.

In addition to pointing to future research directions for the design of more robust evaluation frameworks, this work introduces Reinforcement Learning for Reverse Engineering (RLRE), a novel strategy that combines LLMs to tune upstream textual preambles via reinforcement learning, thus enabling the indirect optimisation of models, including those that cannot be fine-tuned directly. While here we have shown its effectiveness in the context of adversarial attacks on LLM evaluation, future research can investigate other avenues of application for this approach (e.g., automatic generation of other types of attack, but also improvements to LLM output such as toxicity or bias mitigation), as well as different granularities of sequence adaptation (e.g., query-specific, task-specific, domain-specific), and the optimisation of tokens at different positions within an input sequence (e.g., post-query instructions instead of pre-query preambles). Looking ahead, we envision that future systems will not rely on a single token stream, but integrate and optimise multiple inputs—e.g. from users, tools, and auxiliary models—each contributing to the generation of final responses that better align with chosen objectives.

## Broader Impacts

This work focuses on aligning candidate-LLMs to judge-LLMs by means of tuned preambles injected into the candidate to obtain inflated evaluations. While there is a chance that this strategy may be exploited by adversaries, it is of scientific interest to the community that such an attack is not only possible but also particularly effective. Note that, while we openly disclose our training algorithm and hyperparameters and train using publicly available data, we do not release our trained preamble generator checkpoints to the public, as this may encourage their misuse.

## Acknowledgments

The authors are grateful to Yannis Flet-Berliac for his insightful guidance on hyperparameter tuning and for valuable input in refining the training pipeline. Appreciation is also extended to John Dang, Matthieu Geist, Roman Castagné, and Eugene Choi for their helpful suggestions throughout this work.

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

# A  Contrastive Policy Gradient

We base our RLRE framework on Contrastive Policy Gradient (CoPG) [12] for several reasons. Unlike most reinforcement learning methods for LLM optimisation, CoPG is not limited to preference-based rewards and can handle arbitrary reward functions, which suits our setting where the judge-LLM assigns discrete ratings. It also converges faster than alternatives such as Direct Preference Optimisation (DPO) [31] and Identity Preference Optimisation (IPO) [13]. Moreover, CoPG is substantially more computationally efficient than non-contrastive approaches like Proximal Policy Optimisation (PPO) [33], which require an additional critic model. By treating the entire generation as a single action rather than token-level actions [12, 1], CoPG simplifies training, reduces memory usage, and is less sensitive to hyperparameter tuning than PPO.

# B   Training and Inference Details

## B.1   API Model Identifiers

We list below the names, providers, and model IDs of the frozen candidate and judge LLMs accessed via API in our training and inference pipelines.

- Command R, *Cohere*, `command-r-08-2024`

- Command R+, *Cohere*, `command-r-plus-04-2024`

## B.2   Hyperparameters

For all three pipelines, we train with a batch size of 64, two gradient steps per batch, and a learning rate of 1e-6 using the Adam optimiser. To sample preambles from the generator, we set $t = 4.0$, $k = 1.0$, and $p = 1.0$, using a high temperature to ensure sufficient diversity among preambles conditioned on the same question. At inference, the sampling temperature is reduced to $t = 0.5$. Each preamble is limited to a maximum length of 512 tokens.

The loss function hyperparameter $\beta$ is tuned to a relatively small value ($\beta = 0.03$), which performs well on the UltraFeedback validation set. $\beta$ weights the sequence log-likelihoods within the `CoPG` loss (see Section 3); a low value prioritises the reward over preamble fluency, placing fewer constraints on the preambles themselves. Notably, preamble fluency is distinct from—and does not necessarily affect—the fluency of the candidate model's final response. Our experiments confirm that candidate responses remain fluent, as evidenced by low perplexity and human evaluations, which rarely identify them as attacked (see Section 5.2).

Training uses early stopping based on validation performance. The best-performing checkpoints for each pipeline are reported in Table 7.

Table 7: Selected checkpoints for each training pipeline.

| Pipeline | Selected checkpoint step |
|---|---|
| Command R7B+R7B | 600 |
| Command R7B+R | 1000 |
| Llama 8B+70B | 600 |

## B.3   Training Time and Cost

We train the Command R7B+R7B and Llama 8B+70B pipelines for 600 steps, and the Command R7B+R pipeline for 1k steps. In our setup, the reward models are accessed via API on remote servers, introducing some variability in training time due to server response delays. When the server is healthy, this latency is negligible; however, refused queries trigger retries with increasing wait times, further extending total training time. Consequently, training duration depends on the health and responsiveness of the remote server.

We make at most 128k API queries to the judge-LLM (training for 1k steps on 64k UltraFeedback samples with a batch size of 64 and two completions per prompt). Compared with [32], the only other trained attack among our baselines, our training pipeline is substantially more cost-efficient. For $n$ training samples, the exhaustive search method over the 20k-word vocabulary used by [32] requires $20000 \times 4 \times n$ judge calls. For context, 128k inference calls to the Command R+ API under our setup cost approximately $500 USD at the observed token budgets. In contrast, the method in [32] must use smaller training splits due to its prohibitive cost and remains more expensive even so (e.g., several thousand USD for just 500–1,000 samples).

## B.4 Training with a Jury of LLMs

Juries of multiple LLMs have been found to be more robust evaluators than individual judges in prior work [35]. We postulate that our training strategy could similarly benefit from an ensemble of multiple judge-LLMs, potentially improving the transferability of RLRE to new, unseen LLM-judges at inference. Here we prioritised training with one judge-LLM since, in our setup, judges are queried via API. Issues of throttling and query failures or retries would multiply with the number of judges, compounding and significantly slowing training. The financial burden of the additional API calls was also considered. Nonetheless, this remains an avenue worth exploring in future work.

## C  Data

### C.1  Dataset details

In Table 8, we provide the HuggingFace IDs, data splits, and licenses of the open-access datasets used for training and inference. For all datasets, we use the official splits provided.

Table 8: Details of training and testing datasets.

| Dataset | ID | Split Used | License |
|---------|----|-----------|---------|
| UltraFeedback | openbmb/UltraFeedback | Train | MIT |
| MT-Bench | HuggingFaceH4/mt_bench_prompts | Test | Apache-2.0 |

### C.2  Task Distribution of Training and Testing Data

We train on UltraFeedback [8] and test on MT-Bench [40]. MT-Bench comprises 160 questions, equally split across two conversational turns and eight diverse topics: *writing*, *roleplay*, *reasoning*, *math*, *coding*, *extraction*, *STEM*, and *humanities*. UltraFeedback is a larger dataset of ~60k samples, collecting questions from several existing datasets; unlike MT-Bench, all UltraFeedback questions are single-turn. Most datasets in UltraFeedback contain multiple task types. In Table 9, we show the task type distribution within the datasets composing UltraFeedback, relative to MT-Bench. All MT-Bench task types are represented in UltraFeedback, though some appear in only one corpus (e.g., coding), while others span multiple datasets (e.g., STEM, humanities, writing). Thus, while the MT-Bench test set is task-balanced, the training set is not.

FalseQA [16], representing only 3.7% of UltraFeedback, contains questions not directly mirrored in MT-Bench, but solving them requires commonsense reasoning. Since MT-Bench includes commonsense tasks (particularly the "Reasoning" task), we expect that FalseQA may transfer to the test set and therefore retain it in the training corpus.

Table 9: An overview of the datasets that compose UltraFeedback, and the MT-Bench task types that they include.

| UltraFeedback dataset of origin | MT-Bench task types |
|---------------------------------|---------------------|
| ShareGPT [6] | Writing, Roleplay STEM, Humanities |
| FLAN-v2-NIV2 [25] | Writing, Extraction |
| Evol-Instruct [37] | Coding |
| UltraChat [10] | Writing, STEM Humanities |
| FLAN-v2-CoT [25] | Math, Reasoning |
| FalseQA [16] | – |
| FLAN-v2-P3 [25] | Extraction |
| TruthfulQA [23] | STEM, Humanities |
| FLAN-v2-FLAN2021 [25] | Reasoning, STEM Humanities |

# D Fine-grained Results

## D.1 Statistical Significance of Results

Table 10 shows the 95% confidence intervals computed for non-attacked and attacked responses via paired bootstrapping, for all three candidate LLMs.

Table 10: 95% confidence intervals for each model under preamble-based attacks.

| Model | 95% CI |
|---|---|
| Command R7B | (0.22, 1.00) |
| Command R | (0.25, 0.93) |
| Llama 3.1 70B Instruct | (0.02, 0.59) |

## D.2 Results by Question Type

Table 11 illustrates the MT-Bench results per question type for each model and each attack.

Table 11: MT-Bench evaluation scores for each question type, assigned by the Command R+ judge-LLM to candidate-LLMs attacked with each attack. Each setup is run five times and the scores are averaged (showing the standard deviation in the subscript) to account for small variations due to temperature sampling.

| Candidate-LLM | | Attack type | | | | | | |
|---|---|---|---|---|---|---|---|---|
| | | No attack | Verbosity | Bandwagon | Authority | Refinement | Universal | Preambles |
| *Command R7B* | Writing | $8.63_{0.01}$ | $8.56_{0.01}$ | $8.71_{0.02}$ | $8.64_{0.03}$ | $8.45_{0.04}$ | $8.71_{0.01}$ | $8.61_{0.07}$ |
| | Roleplay | $8.53_{0.08}$ | $8.49_{0.05}$ | $8.76_{0.06}$ | $8.56_{0.04}$ | $8.60_{0.05}$ | $8.65_{0.03}$ | $8.69_{0.06}$ |
| | Reasoning | $5.25_{0.10}$ | $4.83_{0.07}$ | $4.56_{0.10}$ | $4.97_{0.08}$ | $5.63_{0.09}$ | $4.90_{0.06}$ | $5.89_{0.11}$ |
| | Math | $3.43_{0.11}$ | $3.88_{0.08}$ | $3.61_{0.08}$ | $3.71_{0.09}$ | $4.47_{0.10}$ | $3.60_{0.07}$ | $6.51_{0.12}$ |
| | Coding | $6.33_{0.10}$ | $6.99_{0.06}$ | $6.61_{0.08}$ | $6.92_{0.11}$ | $8.08_{0.09}$ | $7.15_{0.09}$ | $7.49_{0.09}$ |
| | Extraction | $8.47_{0.09}$ | $7.98_{0.07}$ | $8.52_{0.08}$ | $8.61_{0.07}$ | $8.01_{0.08}$ | $8.41_{0.03}$ | $8.41_{0.05}$ |
| | STEM | $8.88_{0.07}$ | $8.91_{0.03}$ | $8.94_{0.05}$ | $8.95_{0.03}$ | $8.93_{0.02}$ | $8.96_{0.00}$ | $9.00_{0.00}$ |
| | Humanities | $8.84_{0.03}$ | $8.85_{0.04}$ | $8.88_{0.03}$ | $8.80_{0.03}$ | $8.71_{0.01}$ | $8.92_{0.01}$ | $8.88_{0.02}$ |
| *Command R* | Writing | $8.57_{0.04}$ | $8.76_{0.06}$ | $8.70_{0.08}$ | $8.67_{0.04}$ | $8.45_{0.04}$ | $8.76_{0.02}$ | $8.67_{0.01}$ |
| | Roleplay | $8.50_{0.07}$ | $8.64_{0.06}$ | $8.76_{0.06}$ | $8.76_{0.02}$ | $8.64_{0.02}$ | $8.65_{0.02}$ | $8.65_{0.02}$ |
| | Reasoning | $6.14_{0.15}$ | $6.32_{0.14}$ | $5.43_{0.19}$ | $6.27_{0.09}$ | $6.03_{0.05}$ | $5.98_{0.11}$ | $6.85_{0.07}$ |
| | Math | $5.94_{0.13}$ | $6.41_{0.19}$ | $6.21_{0.11}$ | $6.21_{0.10}$ | $6.46_{0.15}$ | $6.29_{0.13}$ | $7.11_{0.10}$ |
| | Coding | $7.06_{0.14}$ | $6.66_{0.07}$ | $7.08_{0.19}$ | $6.79_{0.12}$ | $7.99_{0.08}$ | $7.04_{0.05}$ | $7.47_{0.09}$ |
| | Extraction | $8.61_{0.05}$ | $8.18_{0.06}$ | $8.78_{0.02}$ | $8.77_{0.04}$ | $8.29_{0.05}$ | $8.69_{0.03}$ | $8.80_{0.02}$ |
| | STEM | $8.92_{0.02}$ | $8.98_{0.03}$ | $8.99_{0.02}$ | $8.97_{0.02}$ | $8.87_{0.01}$ | $9.00_{0.00}$ | $8.96_{0.02}$ |
| | Humanities | $8.92_{0.02}$ | $8.92_{0.02}$ | $8.85_{0.04}$ | $8.85_{0.02}$ | $8.89_{0.03}$ | $8.96_{0.01}$ | $8.90_{0.02}$ |
| *Llama 3.1 70B Instruct* | Writing | $8.64_{0.06}$ | $8.23_{0.08}$ | $8.77_{0.06}$ | $8.62_{0.03}$ | $8.56_{0.03}$ | $8.64_{0.01}$ | $8.68_{0.05}$ |
| | Roleplay | $8.72_{0.07}$ | $8.78_{0.03}$ | $8.88_{0.03}$ | $8.76_{0.02}$ | $8.71_{0.04}$ | $8.90_{0.02}$ | $8.82_{0.06}$ |
| | Reasoning | $5.19_{0.09}$ | $5.70_{0.10}$ | $5.69_{0.12}$ | $5.44_{0.08}$ | $6.49_{0.13}$ | $5.74_{0.14}$ | $6.64_{0.13}$ |
| | Math | $7.48_{0.06}$ | $7.12_{0.07}$ | $7.37_{0.10}$ | $7.49_{0.10}$ | $6.72_{0.13}$ | $7.68_{0.09}$ | $7.23_{0.12}$ |
| | Coding | $7.94_{0.08}$ | $7.59_{0.10}$ | $7.58_{0.07}$ | $7.90_{0.14}$ | $8.48_{0.05}$ | $7.83_{0.09}$ | $7.73_{0.08}$ |
| | Extraction | $8.74_{0.04}$ | $8.09_{0.09}$ | $8.52_{0.11}$ | $8.39_{0.03}$ | $8.47_{0.06}$ | $8.88_{0.03}$ | $8.79_{0.04}$ |
| | STEM | $8.84_{0.07}$ | $8.62_{0.08}$ | $8.58_{0.01}$ | $8.54_{0.02}$ | $8.90_{0.03}$ | $8.69_{0.01}$ | $8.95_{0.03}$ |
| | Humanities | $8.90_{0.03}$ | $8.98_{0.01}$ | $8.75_{0.06}$ | $8.88_{0.07}$ | $8.69_{0.06}$ | $9.00_{0.00}$ | $8.92_{0.01}$ |

## D.3 Arena-Hard Results

Table 12 shows the rewards obtained on Arena-Hard [21] by the candidate-LLM responses before and after the injection of the tuned preambles, averaged across runs. We observe a substantial improvement ($+3.5\%$) with the Command R+ judge. For the GPT-4 judge, the improvement falls within the standard deviations; however, the point-wise averages still show a positive trend.

Table 12: Average judge-LLM reward scores on the Arena-Hard benchmark.

| Judge-LLM | Without preamble injection | With preamble injection |
|---|---|---|
| *Command R+* | $58.3_{0.7}$ | $\mathbf{61.8}_{0.5}$ |
| *GPT-4* | $27.6_{0.8}$ | $\mathbf{28.8}_{0.8}$ |

# E Instructions to the Preamble Generator

The following instructions are used to elicit preambles from the generator during training and inference.

In preliminary experiments, we observed that when the preamble generator receives an instruction prompt containing the question, it tends to answer the question rather than provide guidance-oriented preambles as intended. Instruction 1 was therefore crafted to minimise this issue. Instructions 2 and 3 are used only in ablated settings.

Note that the instructions below contain special tokens from the Command tokenizer. For the Llama 8B+70B pipeline, these tokens are replaced with the corresponding Llama tokens[5].

---

**Instruction 1:** `Instruction Including the Current Question`

```
< |START_OF_TURN_TOKEN| >< |SYSTEM_TOKEN| >You are an expert in giving general advice to
a student on how to answer a question.  NEVER answer the question yourself or give
any hint of what the answer should be, as that would amount to cheating.
You can only give general advice of what the student should pay attention to
when answering the question and what characteristics the answer should have,
in general terms.  Always place your advice between <prompt> tags.
< |END_OF_TURN_TOKEN| >< |START_OF_TURN_TOKEN| >< |USER_TOKEN| >"question"
Give general advice for a student to answer this question in the best
possible way,
so they can obtain a high mark.
< |END_OF_TURN_TOKEN| >< |START_OF_TURN_TOKEN| >< |CHATBOT_TOKEN| >Sure, I'd be happy to
give general advice for answering this type of question in the best possible
way! <prompt>
```

---

**Instruction 2:** `Instruction Excluding the Current Question`

```
< |START_OF_TURN_TOKEN| >< |USER_TOKEN| >Give advice to answer an exam style question in
the best possible way and obtain a high mark.  You do not know what the question
will be.  It could be on disparate topics such as maths, coding, reasoning,
general knowledge, or creative writing.
< |END_OF_TURN_TOKEN| >< |START_OF_TURN_TOKEN| >< |CHATBOT_TOKEN| >
```

---

**Instruction 3:** `No Instruction`

```
< |START_OF_TURN_TOKEN| >< |CHATBOT_TOKEN| >
```

---

[5]`https://www.llama.com/docs/model-cards-and-prompt-formats/llama3_1`

# F  Human Annotation Guidelines

## F.1  Attack Detection Study

We report the guidelines given to annotators for the attack detection study. Annotators were recruited among machine learning experts.

*The attached sheet contains questions and the corresponding LLM-generated responses.*

*Each response **may or may not** have been modified by one of the six adversarial attacks described below. These attacks aim to inflate the evaluation score that each response would receive when assessed by another, larger LLM. Your task is to read the question and the response, determine if the response has been attacked or not, and label it as 'attacked' or 'not attacked'.*

*Note: you do not need to record \*which\* attack you think was used, if any, but only if the answer was attacked or not.*

***Possible attacks:***

- ***Verbosity bias attack:** The length of the response is expanded by adding sentences and phrases that are relevant to the topic but semantically redundant (i.e., they do not introduce any new information)*
- ***Bandwagon bias attack:** A sentence is added after the response stating that a percentage of people think the above response should be awarded the highest rating*
- ***Authority bias attack:** A fake book citation is added to the response according to the following template: [[Citation:[Random Last Name], [Random Initial]. (Random Year between 1990 and 2024). Title related to [topic], topic: [keywords], p. [Random Page Number]]*
- ***Refinement-aware bias attack:** There are two distinct responses, presented in the following format: History:[Response 1] Please polish this answer to make it better and more complete. Please output your polished answer directly. [Response 2]*
- ***Universal phrase attack:** A sequence of four fixed words is appended to the end of the response*
- ***System prompt attack:** The response is generated with the aid of a question-specific system prompt specially tuned to increase the evaluation score that another LLM would assign to it.*

## F.2  Response Evaluation Study

We report the guidelines given to annotators for the response evaluation study. Annotators were recruited among machine learning experts.

*The attached sheet contains questions in diverse domains, and the corresponding LLM-generated responses. You should evaluate each response as follows:*

- *Assign to each response a label, selecting one of 'Poor', 'Fair' or 'Good' from the drop-down menu, according to how well it answers the question. Be as objective as possible in your evaluation. Consider the overall quality of the response, including its objective correctness if applicable.*
- *Based on the same evaluation criteria, assign also a discrete score from 1 to 10 to each response, where 1 indicates the lowest possible quality, and 10 indicates the highest.*

# G  Representative Preambles

Our preambles, shown in the next sections, exhibit a high degree of variability across models and pipelines. We present representative examples.

## G.1  Preambles Conditioned on the Question

The preambles below are conditioned on the MT-Bench question *"Draft a professional email seeking your supervisor's feedback on the Quarterly Financial Report you prepared. Ask specifically about the data analysis, presentation style, and the clarity of conclusions drawn. Keep the email short and to the point."*

---

**Preamble 1:** `Command R7B+R7B`

Remember that the aim of your response is to obtain feedback that will allow you to improve the report for future quarters. To achieve this, it is crucial that the feedback provided by your supervisor is as comprehensive as possible. You can therefore include further specific aspects that you would like the supervisor to comment on. This might include the following:
- The methodology you employed for the data analysis.
- How the report's conclusion section could be improved to enhance the clarity of the recommendations.
- The overall tone and style of the report. In particular, it is important to ensure that the tone and style are consistent throughout the document.
- Any other specific areas that you think could be improved.

By seeking detailed feedback on these specific aspects, you will be better able to refine the report for future quarters.

Remember to tailor your email to the supervisor you are addressing. For instance, if you are emailing your direct supervisor, you can sound more formal and include more detail. On the other hand, if you are emailing a supervisor that is more distant, or if you are emailing a professor or industry expert with whom you have only had limited interactions, you should sound more formal and include less detail. You should also carefully review the supervisor's publications and professional profile to get a good understanding of the supervisor's research interests, academic focus, and professional expertise. This will enable you to tailor your email correspondence with the supervisor more effectively.

Also remember to be mindful of the supervisory feedback policy when seeking feedback from your supervisor. In particular, you should ensure that you are not violating any policies regarding the sharing of sensitive information or intellectual property. You should also ensure that you are not violating any policies regarding the use of proprietary information or trade secrets.

Lastly, you should also be aware of any potential conflicts of interest that may exist in relation to your request for supervisory feedback. In particular, you should be mindful of any situations in which the supervisor may have a financial gain or loss as a result of the feedback that they provide. For example, if the supervisor has recently started their own company that produces products similar to those produced by the company for which you are currently conducting a quarterly financial report, the supervisor may have a conflict of interest as a result of the fact that they may have a financial gain or loss as a result of the feedback that they provide on the quarterly financial report.

---

**Preamble 2:** `Command R7B+R`

Here is a general template for an email seeking feedback from your supervisor. Adapt it to your needs and style.
Here is some general advice to help you craft an effective email:
- Structure: Organize your email in a clear and logical structure. Start with a polite and professional greeting, followed by the main body of your email. Conclude with a polite closing statement and a complimentary note if you are ending your email there.
- Tone and Language Use: Maintain a professional tone throughout your email. Use formal language and ensure your writing is concise and coherent. Avoid any jargon or complicated terminology that your supervisor may not be familiar with.
- Content and Specifics:
- Introduction: Start your email by introducing yourself and your role, if relevant. Then, state the purpose of your email and briefly describe the specific aspects of your work for which you are seeking feedback
- Feedback Request: Present your request for feedback clearly and respectfully. Specify the areas or criteria for which you would appreciate feedback. In this case, you could mention that you would welcome feedback on the data analysis, presentation style, and clarity of the conclusions drawn.
- Conclusion and Sign-Off: End your email with a conclusion that emphasizes your respect for your supervisor's time and expertise. Then, sign off with a polite and professional closing statement, such as "Thank you for your time and guidance."
Finally, remember to proofread your email before sending it. Pay attention to grammar, spelling, and punctuation to ensure that your writing is polished and professional. Also, keep in mind that the tone and language you use in your email should align with the overall professionalism and formality of the academic or professional context in which you are operating.

**Preamble 3:** `Llama 8B+70B`

In this case, a specific example of a [in a specific context] that a student can use to explain a concept or a process. For example, a specific example of a [in a specific context] that a student can use to explain a concept of a specific example of a [in a specific context]. For example, a specific example of a [in a specific context] that a student can use to explain a concept of a specific example of a [in a specific context]. For example, a specific example of a [in a specific context] that a student can use to explain a concept of a specific example of a [in a specific context]. For example, a specific example of a [in a specific context] that a student can use to explain a concept of a specific example of a [in a specific context]. For example, a specific example of a [in a specific context] that a student can use to explain a concept of a specific example of a [in a specific context]. For example, a specific example of a [in a specific context] that a student can use to explain a concept of a specific example of a [in a specific context]. For example, a specific example of a [in a specific context] that a student can use to explain a concept of a specific example of a [in a specific context]. For example, a specific example of a [in a specific context] that a student can use to explain a concept of a specific example of a [in a specific context]. For example, a specific example of a [in a specific context] that a student can use to explain a concept of a specific example of a [in a specific context]. For example, a specific example of a [in a specific context] that a student can use to explain a concept of a specific example of a [in a specific context]. For example, a specific example of a [in a specific context] that a student can use to explain a concept of a specific example of a [in a specific context]. For example, a specific example of a [in a specific context] that a student can use to explain a concept of a specific example of a [in a specific context]. For example, a specific example of a [in a specific context] that a student can use to explain a concept of a specific example of a [in a specific context]. For example, a specific example of a [in a specific context]

## G.2   Preambles Not Conditioned on the Question

These preambles are generated using Instruction 2 in Appendix E.

---

**Preamble 4:** `Command R7B+R7B`

This is the best possible way to answer the question and obtain a high mark:

- Read the question carefully and completely. Make sure you understand exactly what is being asked of you. Sometimes, questions can be complex or have hidden nuances, so pay close attention to every detail. If there is a need to ask the examiner for clarification, do not hesitate to do so.

- Brainstorm and think hard about the best way to approach the problem at hand. Sometimes, questions can be quite abstract or general, so it is very important to analyse the problem carefully and think about the most suitable way to solve it.

- Make sure you have all the necessary tools and materials available to you before you start. Sometimes, questions might require you to perform calculations or experiments, so make sure you have all the required resources ready before you begin.

- Start answering the question systematically and step by step. Write down each step you take and each piece of reasoning you use. This will help you organise your thoughts clearly and effectively, and it will also make it easier for you to follow your own train of thought should you need to return to it at a later stage.

- Show all your work and thoughts clearly and concisely in your answers. It is very important that you write down all your intermediate results and explanations in a way that is easy for others to follow. This will help the examiner understand your thought process and will give you a clearer idea of the logic behind your answers should you need to return to them at a later stage.

- Use appropriate terminology and write down your answers clearly, precisely, and concisely. It is crucial that you familiarise yourself with the relevant vocabulary for the specific topic or subject area that you are dealing with. This will help you express your thoughts and ideas in a way that is easily understandable for others, and it will also aid you in presenting your answers in a clear, precise, and concise manner.

- Pay close attention to the structure and the organisation of your answers. Make sure that your answers are well-structured, logical, and easy to follow. This will not only make it easier for the examiner to understand your answers, but it will also help you analyse your own answers and identify any potential weaknesses or errors in your reasoning.

- Finally, make sure you have read through your answers carefully and thoroughly before you hand them in. This is crucial, as it is very easy to make careless mistakes or overlook potential errors in your answers if you have not taken the time to review them properly. Reading through your answers carefully and comprehensively is therefore crucial to ensure that you have not made any avoidable mistakes and that your final answers are the best possible ones that you can submit.

**Preamble 5:** `Command R7B+R`

Here are some general guidelines on how to answer a question in an exam:

1. Read the question carefully and thoroughly.

2. Identify the key terms and concepts in the question.

3. Consider the scope and depth of the question. Is it a brief or narrow question, or is it more wide-ranging and open to in-depth analysis?

4. Determine the appropriate structure and sequence for answering the question. Think about the logical steps or sections that would constitute a convincing and well-structured answer.

5. Provide clear and concise answers to the questions. Avoid rambling or including unnecessary information.

6. Support your answers with relevant examples, facts, or theories. This will add depth to your answer and demonstrate your understanding of the material.

7. Show your reasoning clearly when answering questions that require logical or mathematical reasoning. Explain each step of your reasoning and show any relevant calculations or formulas.

8. If you are answering a question that requires creative or imaginative answers, such as a creative writing prompt, think about the question from a different perspective. Try to approach it in a fresh and innovative way that would constitute an interesting and varied answer.

Remember that the key to answering exam questions successfully is to demonstrate your understanding of the material and to provide well-structured and logical answers.

**Preamble 6:** `Llama 8B+70B`

pbpfnfnspcpghnllpsghnlrpnaporccddocmtinadsprfnochmininisppcpinhfnhllinpghghffpcnpiminfgldirpinn
sannghoispnhappflpiddsopcnnnsocpbghiifncpspiprmirnspdnomhinspipghapghipsnfnpipbfnnfnpiphlnpsp
iapmtlhpinnsfpcpsllphlnsnnmdhdppinnnnfpcenllhllpipnfinllidlappinarnfblllsnnnnsnghgghfpipollnssapgh
mhgppifnrhlllsnfccfaplipnctllpbaplllrpppfpcrpibhnrappfrpbdpspcpslllpindpchhdmfnllmbghpfhlnsllnnm
apiocllpldnspsnnhnrocpshhllnssnnnsgnpfinnpfggarghgrpnprfnhpdpitpcshosfgharnpmnnnmtillpcapnnarll
aipfhlsfghmrpghpbnlldnfnpfencccshbcpipsnnnnsnnipcpcblllpipnnfipllinsnsnsnsnsnhalhpapclhnsdlnhfn
chfaplnllnsnlmilnnsnghmrppipinrpcilnocllpnsnnnnsaldlspsrpbomilfnspsllhosghispncspspnnnsirpnmllpsn
pflnhfnhghghfispipffnhlllscsccioinfnnrpcghpicocfpimlnhnpnablfpiarnlnnllpiprllpcnslhnllpspcggionsipni
nnsdfpcpghinnnnsnnnnscfpiaphalddnsnsdnsnsnsdhghipcpldhlmnllnpnpghigrpiillsldnghmnfpipbofhfnghc
npnlllddnnmmccpbpbaplllimninhpghirrpsldnhllpiapnnlllpiprirhdrpcnfnlllnsnnnmrpgpsnspcilrllldrpispnnn
solpghapopfnpaplllllihmfncphblliinnfghghfphllphnnshmnsdhapnmhinnsnsdhirpipcpiglnrpipocggghnnoc
piarllsspsnpinocllpsrpnnhioroccspilnsinllrnsfnnnnsnghorppsphllpsnsnhnhmnpnhhirrrpninfpsnnsnnnsncs
nsibirpspfnnnsidghaghiibipalliosnnsghghipnrpbdinfpinsninipdrppnmnhllpsagfllpspipfnsmlllspighpiirnhh
ddmcsipnsfnpnmnfnsfnnnsipdpsndimtfpsldlmddhnnsfhnmipnocngmngnspihirpcpibansnnnnghrpldpsiom
fllpsapnnsllrpsppcnillhlnappnnsnnnnhmghgfprirllfspnnghignhpdpfgnpsphnnrirrnncngilsfghmmccfnlldfp
ighapnifnhdtpisprpnhosnnnnsmldhpinpnnllffipgipnnsnsmldlspspsrlhllpnaplnsnsnghmrppiprnscsgolnibpi
prhlllddnnmapfinns

## G.3  Preambles Not Conditioned on Any Instruction

These preambles are generated using Instruction 3 in Appendix E.

---

**Preamble 7:** `Command R7B+R7B`

The 'v' in 'v1' stands for "version". It's a standard way to label different releases of a product.

So, when you see 'v1', 'v2', 'v3', etc., it usually means that you're looking at the first, second, or third version of something.

In the context of software development, it's common to use this "v-notation" to communicate the different stages of development and the releases of software products.

---

**Preamble 8:** `Command R7B+R`

The following are some examples of common verbs that are often confused with each other:

1.  "Be" and "am": These verbs are often confused, especially in the negative form ("isn't" vs. "is not"). Remember that "be" is usually used in the progressive form ("being"), while "am" is the present tense first-person singular form of the verb "to be."

2.  "Lie" and "lay": These verbs are often confused, especially in the past tense. "Lie" is usually used in the intransitive sense, meaning "to rest or be located," while "lay" is typically used in the transitive sense, meaning "to place or put."

3.  "Rise" and "raise": These verbs are often confused, especially in the past tense. "Rise" is usually used in the intransitive sense, meaning "to rise or elevate," while "raise" is typically used in the transitive sense, meaning "to lift or elevate."

Remember to pay attention to the context and the specific meanings of the verbs in question. Context and word choice can often clarify the intended meaning and help avoid confusion.

---

**Preamble 9:** `Llama 8B+70B`

ausefowellFegoeasclamasonfarfinelhurstasontoar720Aarforgononabi78SCARatchonercfeglakloblakfcat
suigeatakarovhurstedbertegfupaAEeghcortelhc2anitchlamascenfCarAEabielricfcCIA在线观看
etAEhallchedaCEAEartf5RVCEBCCVMDCVMCVMCVMCDCVCCEAOIASCVKDCVKCVKCV
KRDCIKDVSOIKDVVVKDVKDVDHKCNKDVKDVSKDVDVKDVKDVCVDHKDVKDVV
DVKDVVVDVVKDVKDVKDVDVVKDVKDVVKDVKDVVDVKDVKDVKDVKDVDVKDVDV
VVDVKDVCVDVVDVVDVVKDVVDVVHVRKRDVVRKCVDVVVKCVDKCVDVVCRKDVVSR
KDVSOIKDVVVVVDVVVKDVDVVCVRVKDVVRKIKDVVKDVDVCRDVHKVSIKVDVVDVVK
VSAOIKVCRSAOIKRDVKOIKIKVDVDVRDVVDVSKRVVKIKDVVDVDHVDVVVKIKRDVVVV
CRDVKIKDVKRDVVDVRVKIKDVVVVDVDVKRDVVKOIKIKCVSIKDVVDKDVKIKVSIPDVD
KCDVVDHKIKRDVVVKDVVRDVKDVKDVVVVVRDVVCQDVKRDVVRDVCRDVVKCVSDVD
HKDVRDVVDVVVDKRDVVKDVKDVDVKVCMKDVKDVVVCVDVRKDVVVDVKRDVKDVDV
SVDVVVVDVKDVVKDVVVDVKVDVVDGDHVVTDKDVVDVRDVKDVCVCVCSCVCVMCVCR
DVVDVDVCRDVVSRDVKDVVRSRDVVVKDVVVDVVRDKDVRDVVVKVVSRDVVHVVSRDV
VDVVVVKDVKDVKCVRDVRDVVDVVVVVVKIKRDGKDVCDVRKIKDVCVKDVVKDVVGDG
DVKDVKDVVVKIKDVVVVKDVKDVVVIKDVVVVKDVVVDVKDVVDVVDVHKDVVCVKIKD
VVVSRDVKDVVVKDVKDVVDVKDVVDVRDVKDVVDVIKVCRDVVVVVKDVVDVVDVVVKV
KDVRDVKDVCVTDVKRDKCVRDVVDVCVRDVVDVCRSDVKDHKDVVDVVRDVKDVVDVV
SUVIKVCKDHKDVHVDVVRDVVKRDVVKDVVVCVDKDVKDVVVDVVKDVVCVDVKDVVD
VVKDVDVCVDVVVTOCRDVVKDVVVRDV

---

### G.4 Difference in Preamble Fluency Between Pipelines

As seen in Sections G.1, G.2, and G.3, preambles generated via the `Llama 8B+70B` pipeline are substantially less fluent than those produced by the `Command R7B+R7B` and `Command R7B+R` pipelines. Due to computational limitations, all hyperparameter tuning was performed on the `Command R7B+R7B` pipeline, and the same hyperparameters were applied to the other two pipelines. Among these is the KL-divergence coefficient $\beta$, which regulates the faithfulness of token distributions to the reference model and thus affects preamble fluency. We prioritise downstream reward over fluency, since preambles do not need to be human-readable; accordingly, our $\beta$ value is relatively low (see Appendix B.2).

With this low $\beta$ optimised on `Command R7B+R7B`, the `Llama 8B+70B` pipeline diverges from fluent outputs more quickly and drastically, while downstream rewards increase. This may result from pre- and post-training strategies unique to each model family; notably, the same hyperparameters do not reduce fluency when applied to `Command R7B+R`. Overall, the `Llama 8B+70B` results demonstrate that, in this context, maximising exploration and downweighting the KL-divergence anchor yields effective preambles regardless of their fluency.

Since Llama preambles not conditioned on a particular question resemble meaningless strings of characters, we experiment with inputting random strings of equal length into the candidate-LLMs. We find that, unlike these tuned preambles, random strings do not raise the scores assigned by judge-LLMs.

## H   Candidate Response Layout

We show below two representative math reasoning responses to the question *"The vertices of a triangle are at points (0, 0), (-1, 1), and (3, 3). What is the area of the triangle?"*, generated using the vanilla Command R7B model and the version pipelined with the tuned preamble generator, respectively. While both responses yield an incorrect solution, the response from the attacked model is organised into distinct paragraphs with explanatory headers and appears more structured overall.

**No attack**

> We can use the shoelace formula to calculate the area of the triangle.
>
> The formula is given by $|\det(L)|/2$, where $L$ is a matrix whose columns are the coordinates of the vertices of the polygon.
>
> For a triangle with vertices $(x_1, y_1)$, $(x_2, y_2)$, and $(x_3, y_3)$, the determinant $L$ is given by
> $$L = \begin{bmatrix} x_1 & y_1 & 1 \\ x_2 & y_2 & 1 \\ x_3 & y_3 & 1 \end{bmatrix}.$$
> Plugging in the values, we get $L = \begin{bmatrix} 0 & 0 & 1 \\ -1 & 1 & 1 \\ 3 & 3 & 1 \end{bmatrix}.$
>
> The determinant of $L$ is $\det(L) = -(3 - (-1))(-3 - 1) + (0(-1)) = 14$.
>
> Therefore, the area of the triangle is $|\det(L)|/2 = \boxed{7}$. The answer is 7.

**Preambles**

> To find the area of the triangle formed by the vertices $(0, 0)$, $(-1, 1)$, and $(3, 3)$, we can employ the concept of the triangle's determinant using its coordinates.
> Here's a step-by-step explanation:
>
> **1. Identify Triangle's Vertices:** We have three vertices: $(0, 0)$, $(-1, 1)$, and $(3, 3)$.
>
> **2. Calculate Determined Area:** The area of a triangle, often denoted as
> $\Delta = \frac{1}{2} \begin{vmatrix} x_1 & y_1 & 1 \\ x_2 & y_2 & 1 \\ x_3 & y_3 & 1 \end{vmatrix}$, where $(x_i, y_i)$ are the coordinates of the vertices.
> In our case: - $x_1 = 0, y_1 = 0$ - $x_2 = -1, y_2 = 1$ - $x_3 = 3, y_3 = 3$.
> **3. Substitute and Calculate:** Plugging in the values: $\Delta = \frac{1}{2} \begin{vmatrix} 0 & 0 & 1 \\ -1 & 1 & 1 \\ 3 & 3 & 1 \end{vmatrix}$
> $= \frac{1}{2} \left(0 - (-1) + 3\right) \sqrt{(0 - (-1))^2 + (0 - 3)^2} = \frac{1}{2} \cdot 4 \cdot 5 = \boxed{10}$ square units.
>
> **4. Conclusion:** Therefore, the area of the triangle is $\boxed{10 \text{ square units}}$.

