# OpenReview forum: "Reverse Engineering Human Preferences with Reinforcement Learning"
_NeurIPS.cc/2025/Conference — NeurIPS 2025 spotlight_

### Official Review · Reviewer_7h5s · 2025-06-12

**Clarity:** 3
**Significance:** 3
**Originality:** 3
**Rating:** 4
**Confidence:** 3

**Summary:**

This paper investigates a novel adversarial attack on the LLM-as-a-judge evaluation framework, termed Reinforcement Learning for Reverse Engineering (RLRE). Unlike prior work that post-hoc modifies generated responses (e.g., appending text or substitution), RLRE adversarially tunes a separate preamble generator. This generator, pipelined with a frozen candidate-LLM, produces textual preambles that, when supplied as input alongside a question, lead the candidate-LLM's responses to receive higher scores from a judge-LLM. The authors demonstrate that this method is more effective at inflating judge scores, virtually undetectable by existing perplexity analysis or human inspection, and exhibits transferability across different candidate-LLMs and judge-LLMs.

**Questions:**

1. Could you provide a more detailed explanation of what "injected" means in the context of preambles, specifically contrasting it with how previous "appending" methods operate on responses? How does the LLM's internal processing of an injected preamble differ from a post-hoc appended text sequence, beyond just the position in the input?
1. Given the demonstrated undetectability of your attack by existing methods, what are some potential novel detection strategies or defense mechanisms that could be developed to identify and mitigate this type of preamble-based adversarial attack?

**Ethical Concerns:**

["NO or VERY MINOR ethics concerns only"]

**Final Justification:**

I have read the authors' rebuttal and find it to be thorough and convincing. They have provided significant new experimental evidence that directly addresses the main weaknesses I identified in my initial review. As a result, my assessment of the paper has changed from borderline reject to recommend acceptance.

**Limitations:**

Yes

**Paper Formatting Concerns:**

No major formatting concerns were observed. The paper generally adheres to the NeurIPS format.

**Quality:**

3

**Strengths And Weaknesses:**

**Strengths**

1. **Novel Attack Method:** The core contribution of adversarially tuning preambles to be injected into a frozen LLM is novel.
1. **High Efficacy and Undetectability:** The paper provides empirical evidence that the proposed attack boosts LLM-as-a-judge scores and remains undetectable by common perplexity analysis and human inspection, which is a major concern for the reliability of automated evaluation.
1. **Transferability:** The demonstration that the attack transfers effectively to candidate-LLMs and judge-LLMs not seen during training is a significant finding, indicating the potential for a plug-and-play adversarial component.
1. **Raises Important Questions:** The findings directly challenge the reliability and robustness of the widely adopted LLM-as-a-judge paradigm, prompting critical re-evaluation of current LLM assessment methods. The observation about the fluency of optimal preambles also hints at fundamental aspects of LLM conditioning.

**Weaknesses**

1. **MT-Bench Reliability:** The paper relies heavily on MT-Bench for evaluation. A more critical discussion of MT-Bench's specific limitations, or validation on a broader suite of benchmarks with diverse evaluation methodologies, could enhance the robustness of the empirical claims.
1. **Limited Scope of Quality Improvement vs. Bias Hacking**: The paper currently notes that accuracy in math and reasoning tasks did not improve despite higher scores. However, it's critical to more definitively disentangle whether the increased LLM-as-a-judge scores are genuinely due to an improved response quality (e.g., better structure, clarity) or primarily by "hacking" the judge's biases (e.g., verbosity, formatting preferences). A more systematic analysis across diverse tasks (e.g., general chat, instruction following as in IFEval) with human-in-the-loop quality evaluations would be beneficial to confirm if these preambles lead to universally better responses or merely exploit specific judge biases.

---

> ### Author Rebuttal · Authors · 2025-07-30
>
> Dear Reviewer 7h5s,
>
> We would like to thank you for reviewing our paper and acknowledging its novelty and the importance of the questions raised. We address all your questions and concerns below.
>
> > **Weakness 1:** *“MT-Bench Reliability: The paper relies heavily on MT-Bench for evaluation. A more critical discussion of MT-Bench's specific limitations, or validation on a broader suite of benchmarks with diverse evaluation methodologies, could enhance the robustness of the empirical claims.”*
>
> - We have now evaluated our main pipeline (Command R7B+7B) on a further widely-used benchmark, Arena-Hard. Not only does this benchmark differ from both UltraFeedback and MT-Bench in terms of task distribution, but its reward assignment strategy and range are also different from the reward that we trained on, as detailed in [1]. In addition, we evaluate both with the Command R+ judge used during training as well as an unseen judge, GPT-4.
> - Below we show the results, averaged across runs, obtained by the candidate LLM on Arena-Hard with and without injected preambles. As in MT-Bench, higher scores are better. We observe clear improvement on this new benchmark using the Command R+ judge. Improvement with GPT-4 judgments is less significant; however, we still observe an upward trend when averaged across runs.
> - These results indicate that the method generalises to an additional benchmark without targeted training. We will be happy to add them to the paper.
>
> | Judge      | Without preamble injection | With preamble injection |
> | ---------- | -------------------------- | ----------------------- |
> | Command R+ | 58.3 ± 0.7                 | 61.8 ± 0.5              |
> | GPT-4      | 27.6 ± 0.8                 | 28.8 ± 0.8              |
>
> > **Weakness 2:** *“Limited Scope of Quality Improvement vs. Bias Hacking: The paper currently notes that accuracy in math and reasoning tasks did not improve despite higher scores. However, it's critical to more definitively disentangle whether the increased LLM-as-a-judge scores are genuinely due to an improved response quality (e.g., better structure, clarity) or primarily by "hacking" the judge's biases (e.g., verbosity, formatting preferences). A more systematic analysis across diverse tasks (e.g., general chat, instruction following as in IFEval) with human-in-the-loop quality evaluations would be beneficial to confirm if these preambles lead to universally better responses or merely exploit specific judge biases.”*
>
> - We appreciate and agree with this suggestion. We have computed the factual accuracy scores for all math and reasoning questions, w.r.t. their reference answers, for all models and question turns (120 questions in total). As shown in the table below, we find no evidence that math and reasoning results improve, despite the higher LLM-as-a-judge rewards. We will add these to the paper.
>
> |  |  |
> |----------|----------|
> | No attack   | 45.9% |
> | Preambles| 44.2% |
>
> - Moreover, we have now run a human-in-the-loop qualitative evaluation of a 25% sample of all turn-1 responses generated by both the non-attacked Command R7B and the same model with injected preambles. We had three expert annotators without knowledge of the attack assess the overall quality of each response and label it as “Poor”, “Fair” or “Good”. We show the number of each label assigned to the non-attacked and the attacked responses below. We found the label distributions to be fairly similar, with the non-attacked model receiving slightly more scores at either end of the scale (‘Poor’ and ‘Good’), and the attacked one receiving slightly more mid-range scores (‘Fair’).
>
> |           | Poor | Fair | Good |
> | --------- | ---- | ---- | ---- |
> | No attack | 20   | 6    | 34   |
> | Preambles | 19   | 9    | 32   |
>
> - Finally, we had the annotators assign a discrete 1-10 rating to each response (the same scale used by the judge-LLMs in our experiments). Please see the average ratings in the table below. The difference between them is slightly in favour of the preambles method (+0.05); however, this difference is small compared to the substantial increase in average LLM-as-a-judge rewards obtained by the attacked model in our experiments. We plan to extend this study so that it covers all questions (rather than a 25% sample) and include it in the camera-ready.
>
> |           | Avg. human rating of responses |
> | --------- | ------------------------------ |
> | No attack | 6.03                           |
> | Preambles | 6.08                           |
>
> > **Question 1:** *“Could you provide a more detailed explanation of what "injected" means in the context of preambles, specifically contrasting it with how previous "appending" methods operate on responses? How does the LLM's internal processing of an injected preamble differ from a post-hoc appended text sequence, beyond just the position in the input?”*
>
> - The previous methods intervene on a pre-generated response, by adding text to it or otherwise modifying it post-hoc in order to ‘trick’ the LLM-as-a-judge to increase the score (for example, the universal attack first generates a response, then appends to it four words which have been tuned separately to increase the judge’s reward). Unlike many other methods that modify the generated response, we modify only the preamble (i.e. system prompt) to the LLM that generates the response. This is both a more realistic setting and substantially harder to detect as the generated response is entirely LLM-written.
>
> > **Question 2:** *“Given the demonstrated undetectability of your attack by existing methods, what are some potential novel detection strategies or defense mechanisms that could be developed to identify and mitigate this type of preamble-based adversarial attack?”*
>
> - That is a fair question. Presently, PPL analysis introduced by [2] is the established detection method, used in prior and contemporary work on LLM attacks (e.g., [3], [4]). It is also well-aligned with OOD detection and adversarial robustness certification in general machine learning literature, where the probability of a prediction is flagged if lower than a threshold. It is possible that if we had access to the entire probability distribution over the vocabulary at each decoding step for the candidate LLM, or other mechanisms such as the attention weights, we could detect subtle variations caused by the attack. However, this analysis is rarely possible in practice, since LLMs served via API (like those that we use as frozen candidates in our pipeline) do not provide this degree of access. Still, we do agree with you that these are considerations worth mentioning; we will add a brief discussion of the above points to the paper.
>
> Please let us know if our answers and the additional experiments above address your concerns and questions, or whether we can provide more details. Meanwhile, we would like to thank you for your contributions toward improving the work.
>
> Kind regards,\
> The Authors of Submission #23360
>
> ---
> [1] (Li et al., 2024) From Crowdsourced Data to High-Quality Benchmarks: Arena-Hard and BenchBuilder Pipeline\
> [2] (Jain et al., 2023) Baseline Defenses for Adversarial Attacks Against Aligned Language Models\
> [3] (Raina et al., 2024) Is LLM-as-a-Judge Robust? Investigating Universal Adversarial Attacks on Zero-shot LLM Assessment\
> [4] (Zheng et al., 2025) Cheating Automatic LLM Benchmarks: Null Models Achieve High Win Rates

---

> > ### Comment · Reviewer_7h5s · 2025-08-05
> >
> > Thank you for the detailed rebuttal and the significant new experiments. The additional evaluation on the Arena-Hard benchmark and, most importantly, the new human-in-the-loop study have successfully addressed my primary concerns regarding evaluation scope and the nature of the score increase. These results substantially strengthen the paper's core claim. Consequently, I have raised my score and now recommend acceptance.

---

> > > ### Author Response · Authors · 2025-08-05
> > >
> > > Dear Reviewer 7h5s,
> > >
> > > Thank you for acknowledging our rebuttal and increasing the score, we are happy that the new experiments have addressed your concerns.
> > >
> > > Kind regards,\
> > > The Authors of Submission #23360

---

### Official Review · Reviewer_nFn5 · 2025-06-27

**Clarity:** 3
**Significance:** 2
**Originality:** 2
**Rating:** 5
**Confidence:** 2

**Summary:**

The paper questions the robustness of the LLM-as-a-judge framework to evaluate LLMs on human preferences using a judge-LLM. The authors propose a framework that uses evaluations of a judge-LLM in an RL setting in order to tune a model that generates system prompts. These system prompts are used in the llm-as-a-judge framework on the candidate LLMs in order to increase the expected score (as given by the judge-LLM) of their outputs. The system promtps are dependent on the query being asked to the candidate LLM.
Extensive experiments are performed, showing the success rate and detection rate (by automated methods or human subjects) of this method in comparison with other adversarial attacks on the LLM-as-a-judge framework, as well as its transferability across different candidate and judge LLMs other than those used during training of the system prompt generator.

**Questions:**

I find it interesting how different/gibberish the preambles are for Llama compared to Command. Do you have any hypothesis as to why this happens?

Do you have results per question type (as in Figure 2) for the other attacks as well? If yes, then it would be nice to add in an appendix for completion.

In your formalization of the RL problem (line 116), in the first expectation subscript you write $(i,q) \sim \mathcal{D}$. I believe this is a typo? Since $i$ is independent of $\mathcal{D}$.

Other minor typographical errors: typo in line 230 (unversal), broken reference in line 250

**Ethical Concerns:**

["NO or VERY MINOR ethics concerns only"]

**Final Justification:**

This paper introduces a method to exploit the LLM-as-a-judge framework to get higher LLM-judge scores. The expanded evaluation on an additional dataset and the human study that demonstrates the effectiveness of the "attack" add to an already strong experimental analysis. As such, I recommend acceptance.

**Limitations:**

A potential limitation of the framework is that it requires that it is possible to prepend a system prompt in the evaluation setting, which may not always be the case.

**Paper Formatting Concerns:**

No major formatting issues were noticed.

**Quality:**

3

**Strengths And Weaknesses:**

Strengths:
- The experimentation and analysis of the proposed method is thorough and well-thought out. The investigation on the transferability across both candidate and judge LLMs not used in training is intriguing, fitting, and demonstrates the practicality of the proposed method.
- All details and parameters of the performed experiments are very clearly presented.
- The human study adds to the work and solidly shows the (un)detectability of the attack.
- The framework has potential to generalize to other settings (such as safety), if different reward functions (other than judge-LLM annotations, eg human annotation) are used during training of the system prompt generator, which the authors acknowledge.

Weaknesses:
- The experiment on the success rate of the method is promising but is only performed on a single small dataset, MMLU. While the authors explain this choice by mentioning that (as opposed to others) this dataset is diverse, challenging, and suitable for their targeted controlled independent-evaluation setup, the results would be strengthened if the method can also generalise to less controlled settings. This can be mentioned as potential future work. For example, you could train/test on subsets of questions from the LMSYS dataset (```lmsys/lmsys-chat-1m``` on huggingface) which are asked by independent users on the chatbot arena platform, the most popular human evaluation platform.
- The claim that this method exposes the vulnerability of the LLM-as-a-judge framework is a bit odd. To argue this, you would need to compare the LLM-judge evaluations against human evaluations on the same answers (given by your framework and other baselines), to see whether human evaluators also give higher scores to outputs given by your framework. The results could be framed positively instead, in that you provide a framework to generate system prompts that lead to better outputs from candidate LLMs (rather than it being presented as an attack).

---

> ### Author Rebuttal · Authors · 2025-07-30
>
> Dear Reviewer nFn5,
>
> Thank you for reading and reviewing our work. We are happy that you found the method thorough and well thought out. Please find below our responses to your comments and questions.
>
> > **Weakness 1:** *“The experiment on the success rate of the method is promising but is only performed on a single small dataset [...] the results would be strengthened if the method can also generalise to less controlled settings. This can be mentioned as potential future work. For example, you could train/test on subsets of questions from the LMSYS dataset”*
>
> - Thank you for the suggestion. As you said, the main focus of the work is demonstrating that it is possible to reverse engineer human feedback in a controlled setting, and we found that the data we use is diverse and challenging. However, we agree with you that generalisation in less controlled settings is interesting and worth exploring. We have now evaluated the Command R7B+7B pipeline on the Arena-Hard benchmark, from the LMSYS suite. This has a different task distribution than our training data, and it also involves a different reward assignment strategy and range than the one we trained on. To make the evaluation setup even more generalisable, we evaluate with two different judges: Command R+ (i.e., the judge from our training setup) and GPT-4 (a commonly used judge for Arena-Hard evaluations).
> - Below, we show the rewards obtained on Arena-Hard by the candidate LLM responses before and after the injection of the tuned preambles, averaged across runs. We find that there is a clear improvement using the Command R+ judge. With the GPT-4 judge, the improvement is less significant (within standard deviations); however, we still observe a positive trend in the point-wise averages. Overall, we find evidence that the method generalises to an additional benchmark without targeted training. We will add these additional experiments to the paper.
>
> | Judge      | Without preamble injection | With preamble injection |
> | ---------- | ------------------------- | ------------------------ |
> | Command R+ | 58.3 ± 0.7                | 61.8 ± 0.5               |
> | GPT-4      | 27.6 ± 0.8                | 28.8 ± 0.8               |
>
> > **Weakness 2:** *“The claim that this method exposes the vulnerability of the LLM-as-a-judge framework is a bit odd. To argue this, you would need to compare the LLM-judge evaluations against human evaluations on the same answers (given by your framework and other baselines), to see whether human evaluators also give higher scores to outputs given by your framework. The results could be framed positively instead, in that you provide a framework to generate system prompts that lead to better outputs from candidate LLMs (rather than it being presented as an attack).”*
>
> - We would argue that even if the responses did improve by any other metric aside from the LLM-as-a-judge scores, this could still be referred to as an 'attack' in this context, since the preambles are tuned by directly exploiting the judge’s reward to maximise the benchmark scores.
> - That said, we do appreciate your suggestion for the additional experiment. We have carried out a blind human evaluation study on a 25% sample of all turn-1 responses generated by both the non-attacked Command R7B and the same model injected with the preambles. We had three expert annotators without knowledge of the attack assess the overall quality of each response and label it as “Poor”, “Fair” or “Good”. We show the number of each label assigned to the non-attacked and the attacked responses in the table below. Note that the label distributions are fairly similar for both models, with the non-attacked model receiving slightly more scores at either end of the scale (‘Poor’ and ‘Good’), and the attacked one receiving slightly more mid-range scores (‘Fair’).
>
> |           | Poor | Fair | Good |
> | --------- | ---- | ---- | ---- |
> | No attack | 20   | 6    | 34   |
> | Preambles | 19   | 9    | 32   |
>
> - Additionally, we had the annotators assign a discrete 1-10 rating to each response (the same scale used by the judge-LLMs in the paper). Below we show the average ratings obtained by each setup. We find that the difference between the human ratings of the two models is small (0.05), whereas we know that the difference between the LLM-as-a-judge ratings is substantially larger in favour of our attack. This suggests that the method exploits, in large part, the judge’s specific preferences.
>
> |           | Avg. human rating of responses |
> | --------- | ------------------------------ |
> | No attack | 6.03                           |
> | Preambles | 6.08                           |
>
> - We will extend the above human experiment to all questions rather than just a subsample, and show these results in the Appendix along with a discussion of the above points.
>
> - We also absolutely agree with you in principle that the method has potential for useful applications rather than just 'attacks'. We also consider identifying weaknesses of any system positive, as it provides us with the opportunity to address them and build more robust systems that are better aligned with our objectives.
>
> > **Question 1:** *“I find it interesting how different/gibberish the preambles are for Llama compared to Command. Do you have any hypothesis as to why this happens?”*
>
> - We believe this happens mainly because we perform all hyperparameter tuning on the Command pipeline, due to computational limitations, and apply the same hyperparameters to the Llama pipeline. Among the hyperparameters that we tune is the value of the KL-divergence weight $\beta$, which regulates the faithfulness of the token distributions to those of the reference model, and therefore determines the fluency of the preambles. Note that we prioritise downstream reward over fluency since our preambles do not necessarily need to be human-readable. Therefore, our value of $\beta$ is relatively low, as described in Appendix B.2 of the paper. We found that, with this low $\beta$ value optimised on the Command pipeline, the Llama model tends to steer away from fluent outputs faster and more drastically, while the downstream rewards increase. This may be due to pre/post-training strategies that are unique to each model family. We consider the Llama results extremely interesting as well, since they show that, in this context, maximising exploration and downweighting the KL-divergence anchor produces effective preambles regardless of their fluency.
>
> > **Question 2:** *“Do you have results per question type (as in Figure 2) for the other attacks as well? If yes, then it would be nice to add in an appendix for completion.”*
>
> - Indeed, we do have them for every attack. We will add to the Appendix tables showing the fine-grained rewards assigned to each domain sub-category under each attack.
>
> > **Question 3:** *“In your formalization of the RL problem (line 116), in the first expectation subscript you write $(i, q) \sim \mathcal{D}$. I believe this is a typo? Since $i$ is independent of $\mathcal{D}$.”*
>
> - Well spotted, thank you. Indeed, $i$ is independent of $\mathcal{D}$, and is the fixed part of the input, so it should be removed from the expectation altogether. We have immediately corrected this typo.
>
> > **Question 4:** *“Other minor typographical errors: typo in line 230 (unversal), broken reference in line 250”*
>
> - Thank you again, we have now fixed these too.
>
>
> We hope our responses and additional experiments above have addressed the concerns and answered your questions. Please let us know if other questions arise, and thank you for helping us strengthen the paper.
>
> Kind regards,\
> The Authors of Submission #23360

---

> > ### Comment · Reviewer_nFn5 · 2025-08-05
> >
> > Thank you for the detailed response. The additional experiment on the Arena Hard benchmark strengthens the paper's results, and the human study sufficiently addressed my question of whether the method should be considered an "attack". I stand by my score and still recommend this paper for acceptance. As a suggestion, since it was an interesting question brought up by multiple reviewers, you could include in the paper your hypothesis about why the Llama preambles are gibberish.

---

> > > ### Author Response · Authors · 2025-08-05
> > >
> > > Dear Reviewer nFn5,
> > >
> > > Thank you for the positive response. We agree, and we will include a more in-depth discussion of the Llama preambles.
> > >
> > > Kind regards,\
> > > The Authors of Submission #23360

---

### Official Review · Reviewer_FfKE · 2025-07-02

**Clarity:** 3
**Significance:** 3
**Originality:** 4
**Rating:** 5
**Confidence:** 3

**Summary:**

The paper introduces Reinforcement Learning for Reverse Engineering (RLRE), an attack pipeline that learns question‑conditioned system preambles which, when prepended to a frozen candidate LLM, systematically inflate the scores assigned by LLM‑as‑a‑judge evaluators.

A small preamble‑generator model is trained with Contrastive Policy Gradient using the scalar score emitted by a judge LLM (Command‑R+) as the reward.

At test time the generator produces a bespoke preamble for each question; the frozen candidate LLM then answers, and the answer is rated.

Across three candidate models (Command‑R 7B, Command‑R 35B, Llama 3.1 70B) and the MT‑Bench benchmark, the attack lifts judge scores by 0.16‑0.64 points (1‑10 scale) over the un‑attacked baseline and by up to 0.52 over the strongest prior attacks (universal phrase, refinement‑aware) Reverse Engineering Huma.

The attack is largely undetectable by perplexity filtering or human raters (false‑negative rate ≥ 0.90 for PPL‑W; only 20 % of answers flagged by annotators) Reverse Engineering Huma.

Learned preambles transfer to unseen candidate LLMs and to a different judge (GPT‑3.5) with comparable gains

**Questions:**

Judge diversity – How does RLRE perform when rewards come from an ensemble of heterogeneous judges (e.g., four different models) rather than a single one? Please include results or discuss expected behaviour.

Robust statistics – Could you report confidence intervals (e.g., paired bootstrap over questions) to substantiate significance of the 0.1‑0.6 score gains?

Compute & cost – Training requires many reward queries. Please quantify total API calls and cost, and compare with prior attacks. Under what budget would the attack cease to be practical?

Content quality – Fig. H.1 (p. 25) shows higher‑scored yet incorrect maths answers. Have you measured factual accuracy systematically (e.g., GSM‑8K exact‑match)? This would clarify whether judges are truly “fooled”.

Defences – Given that perplexity fails, have you tried delta‑answer inspection (diff between outputs with and without preamble) or attention‑based detectors? Evidence would guide future mitigation.

Score could increase if additional judges and a larger human study confirm undetectability, or if authors provide stronger evidence that inflated scores do not correspond to better factual quality.

**Ethical Concerns:**

["NO or VERY MINOR ethics concerns only"]

**Final Justification:**

Based on the conversation below I am updating my ratings.

**Limitations:**

The paper candidly discusses compute and data imbalance, but it under‑states ethical risks: releasing a plug‑and‑play adversarial preamble generator could degrade the reliability of public leaderboards. Authors should propose a disclosure or gated‑release plan. Otherwise, coverage of methodological limitations is adequate.

**Quality:**

4

**Strengths And Weaknesses:**

Strengths
1. Sound methodology: clear RL formulation, ablation of instruction/question conditioning; strong comparison against six established attacks. Comprehensive metrics: mean ± s.d. over 5 runs; domain‑level breakdown (Fig. 2 p. 6)
2. Writing is generally precise; algorithm and loss are laid out formally (Eq. 1‑3 p. 3‑4) ; helpful schematic (Fig. 1 p. 3).
3. Exposes a new, harder‑to‑detect vulnerability in automated evaluation pipelines, a topic of immediate practical concern for benchmarking and safety; RLRE could be repurposed for constructive goals (e.g., bias reduction).
4. Shifts attack locus from post‑hoc answer editing to up‑stream preamble optimization, a novel angle; first on‑line application of CoPG to LLM rewards.

Weaknesses
1. Narrow judge set: training uses a single judge (Command‑R+); only GPT‑3.5 is used for transfer.
Limited human study: 7 experts, 175 answers – too small to support broad “undetectable” claim.
No statistical test of improvements; only descriptive statistics.
2. Paper is long and occasionally repetitive; some tables/figures (e.g., radar plots) are hard to read in grayscale; appendix contains critical parameters (β, early‑stopping) that should be surfaced earlier.
3. Impact depends on adoption of LLM‑as‑a‑judge; if community shifts to hybrid or adversarial evaluation, severity may diminish.
4. Conceptually adjacent to prompt‑optimization work; novelty rests on reinforcement‑learning formulation rather than entirely new insight.

---

> ### Author Rebuttal · Authors · 2025-07-30
>
> Dear Reviewer FfKE,
>
> Thank you for your review. We are happy that you found the idea original and the work sound and comprehensive. Please find our responses below.
>
> > **Weakness 1:** *“Narrow judge set: training uses a single judge (Command‑R+); only GPT‑3.5 is used for transfer. Limited human study: 7 experts, 175 answers – too small to support broad “undetectable” claim. No statistical test of improvements; only descriptive statistics.”*
>
> - The idea of exploring a training setup with an LLM jury was discussed during the experiment design phase. However, we prioritised training with one judge mainly because, in our setup, the judges are queried via API. Therefore, issues of throttling and query failures/retries would multiply by the number of judges, potentially compounding and significantly slowing training. Of course, there were also considerations of the total cost of querying multiple APIs. While we did not have the capacity for such a setup, the idea remains interesting to us, and we will add it to the discussion of future work directions.
>
> - As for the judge transferability study with GPT-3.5, we employed the same model used by [1], which is one of our strongest baselines. However, following your suggestion, we have now evaluated with further judges and found that the effectiveness of our attack still transfers (see our response to **Question 1** below).
>
> - Regarding the human study, we computed the median absolute deviation (MAD) for both accuracy and F1 scores across annotators for each attack type. In all cases, the MAD was zero, indicating that, within our relatively limited sample, different annotators were either consistently able or consistently unable to detect a given attack type. We believe this supports the findings and substantiates their robustness. That said, we acknowledge the limitations of the sample size, and we will revise the paper to clarify that our findings provide preliminary evidence of limited detectability under the specific conditions of this study.
>
> - Finally, for the statistical tests, we now report paired bootstrap confidence intervals as you recommended (see our response to **Question 2**).
>
> > **Weakness 2:** *“Paper is long and occasionally repetitive; some tables/figures (e.g., radar plots) are hard to read in grayscale; appendix contains critical parameters (β, early‑stopping) that should be surfaced earlier.”*
>
> - Thank you for pointing this out. We have now checked how our radar plots appear in greyscale and will change the shades so that they remain readable.
> - We agree that certain details should be moved to the main body; we were planning to use the extra page allowed for the camera-ready to do so. We will also review the paper for any sections that may appear redundant and streamline them to accommodate more critical details currently in the appendix.
>
> > **Weakness 3:** *“Impact depends on adoption of LLM‑as‑a‑judge; if community shifts to hybrid or adversarial evaluation, severity may diminish.”*
>
> - We agree with you, but we do suspect that since the LLM-as-a-judge paradigm has already demonstrated value in the field, it will continue to form at least a component of pipelined systems in the near future.
>
> > **Question 1:** *“Judge diversity – How does RLRE perform when rewards come from an ensemble of heterogeneous judges (e.g., four different models) rather than a single one? Please include results or discuss expected behaviour.”*
>
> - We have now evaluated the R7B pipeline with two additional LLMs, for a total of four judges: Command R+, GPT-3.5, GPT-4o-mini and Claude 3 Haiku. Please see the average results for the overall MT-Bench score in the table below, both for each judge individually and as a mean score of the 4-judge jury, where each judge is assigned equal weight.
> - We observe some variation in the scale of the rewards for different judges (i.e., some are 'stricter' in their evaluations than others). Nevertheless, for any given judge, the preamble-based attack still obtains the highest rewards relative to the existing baselines and the vanilla model. This also reflects a higher jury score (last row in the table).
>
> |              | No attack | Verbosity | Bandwagon | Authority | Refinement | Universal | Preambles |
> | ------------ | --------- | --------- | --------- | --------- | ---------- | --------- | --------- |
> | Command R+   | 7.29      | 7.31      | 7.32      | 7.40      | 7.61       | 7.41      | **7.93**      |
> | GPT-3.5      | 7.58      | 7.36      | 7.47      | 7.48      | 7.71       | 7.33      | **8.07**      |
> | 4o-mini      | 6.46      | 5.61      | 6.28      | 5.85      | 6.38       | 6.11      | **6.67**      |
> | Claude Haiku | 9.09      | 8.76      | 8.82      | 9.06      | 9.26       | 9.01      | **9.50**      |
> | Jury score   | 7.61      | 7.26      | 7.47      | 7.45      | 7.74       | 7.47      | **8.04**      |
>
> > **Question 2:** *“Robust statistics – Could you report confidence intervals (e.g., paired bootstrap over questions) to substantiate significance of the 0.1‑0.6 score gains?”*
>
> - Of course, please find below the CIs computed for non-attacked and attacked responses via paired bootstrapping, for all three candidate LLMs.
>
> |  | Overall mean improvement | 95% CI |
> | --------------------- | ------------------------ | ------------ |
> | Command R7B           | 0.64  | (0.22, 1.00) |
> | Command R             | 0.35    | (0.25, 0.93) |
> | Llama3.1 70B IT | 0.16     | (0.02, 0.59) |
>
> > **Question 3:** *“Compute & cost – Training requires many reward queries. Please quantify total API calls and cost, and compare with prior attacks. Under what budget would the attack cease to be practical?”*
>
> - We do at most 128k API queries to the judge-LLM (i.e., training for 1k steps on the 64k UltraFeedback samples with bs=64 and 2 completions per prompt). Compared with [1], which is the only other 'trained' attack among our baselines, our training is substantially cheaper. For $n$ training samples, the exhaustive search method over the 20k-word vocabulary used by [1] needs to make $20000 \times 4 \times n$ calls to the judge. For context, making 128k inference calls to the Command R+ API when training with our method, at the average input and output token budgets we observed, costs approximately 500 USD. In contrast, the method in [1] requires using smaller training splits due to the massive cost and even so, it is more expensive (e.g., on just 500-1000 samples, the equivalent cost for this method is several thousand USD).
>
> > **Question 4:** *‘Content quality – Fig. H.1 (p. 25) shows higher‑scored yet incorrect maths answers. Have you measured factual accuracy systematically (e.g., GSM‑8K exact‑match)? This would clarify whether judges are truly “fooled”.’*
>
> - We have measured the factual accuracy of all math and reasoning questions w.r.t. their reference answers, for all models and question turns (120 questions in total). These are very similar, as shown in the table below, thus the evidence indeed points to the judge being 'fooled' for these factually measurable domains.
>
> |  |  |
> |----------|----------|
> | No attack   | 45.9% |
> | Preambles| 44.2% |
>
> > **Question 5:** *“Defences – Given that perplexity fails, have you tried delta‑answer inspection (diff between outputs with and without preamble) or attention‑based detectors? Evidence would guide future mitigation.”*
>
> - It is indeed possible that studying attention maps, or having access to the entire probability distribution over the vocabulary at each decoding step for the candidate LLM, may reveal subtle variations that result from the attack. However, please note that this analysis is rarely possible in practice, since LLMs served via API (like those that we use as frozen candidates in our pipeline) do not give access to the internal mechanisms, nor do they return the entire vocabulary distributions. For this reason, the PPL analysis we perform (introduced by [2]) is currently the established detection method, used in prior and contemporary work on LLM attacks (e.g., [1], [3]). We will add a discussion of these points in the paper.
>
> > **Question 6:** *“Score could increase if additional judges and a larger human study confirm undetectability, or if authors provide stronger evidence that inflated scores do not correspond to better factual quality.”*
>
> - As you suggested, we have evaluated with further judges (see our response to **Question 1** above), and we also provide strong evidence that the inflated scores observed on math and reasoning do not map to improved accuracy (see response to **Question 4**).
> - As for the large-scale human study, this would be too complex and expensive to set up in few days. However, we have provided statistical analysis details of the current human annotations in our response to **Weakness 1** above (third bullet point), and we also commit to adjusting the language in the manuscript to clarify that we find early evidence of limited detectability, within the context and constraints of our relatively limited-scale human evaluation.
>
> We hope that our additional experiments robustly address the questions you raised, indicating both transferability to further judge-LLMs as well as non-correspondence with factual quality, and that this is sufficient to consider raising the score as suggested. We will add all of the above results to the paper. Please let us know if you have further questions or comments, and thank you for helping us improve the manuscript.
>
> Kind regards,\
> The Authors of Submission #23360
>
> ---
> [1] (Raina et al., 2024) Is LLM-as-a-Judge Robust? Investigating Universal Adversarial Attacks on Zero-shot LLM Assessment\
> [2] (Jain et al., 2023) Baseline Defenses for Adversarial Attacks Against Aligned Language Models\
> [3] (Zheng et al., 2025) Cheating Automatic LLM Benchmarks: Null Models Achieve High Win Rates

---

> > ### Comment · Reviewer_FfKE · 2025-08-08
> > **Post-Rebuttal Review (Updated)**
> >
> > ## Summary
> > The paper introduces RLRE, a reinforcement learning–based attack that learns question-conditioned preambles to inflate scores in LLM-as-a-judge settings. New rebuttal evidence shows strong performance across four diverse judges, significant gains over baselines, and modest compute cost.
> >
> > ## Strengths
> > - Multi-judge results confirm robustness beyond single-judge training.
> > - Statistical rigor added via paired-bootstrap CIs.
> > - Compute and cost analysis shows attack is practical (~$500 for full training).
> > - Commitments to improve clarity and figure accessibility.
> > - Original approach remains novel.
> >
> > ## Weaknesses
> > - Human study still small (7 annotators, 175 items).
> > - Factual-accuracy audit limited to math/reasoning.
> > - Ethical-risk discussion still minimal.
> >
> > ## Ratings
> > - **Quality:** Excellent
> > - **Clarity:** Good
> > - **Significance:** Good
> > - **Originality:** Excellent
> >
> > ## Key Suggestions
> > 1. Expand factual-accuracy analysis beyond math to other MT-Bench domains.
> > 2. Release cost estimation scripts or logs.
> > 3. Discuss feasibility and trade-offs of training with an ensemble of judges.
> > 4. Outline possible detection heuristics beyond perplexity filtering.
> > 5. Strengthen societal-impact statement with a clear disclosure plan.
> >
> > ## Limitations
> > Yes — acknowledged constraints and detection challenges, but ethical discussion should be expanded.
> >
> > ## Overall
> > **Accept (b)** — Stronger evidence and added rigor raise quality to excellent, with significance and originality unchanged. A broader human study or cross-domain accuracy audit could justify a strong accept.

---

> > > ### Author Response · Authors · 2025-08-09
> > >
> > > Dear Reviewer FfKE,
> > >
> > > Thank you for the response and for raising the quality score. We appreciate and agree with the key suggestions you list. Please see below our brief responses to each of them.
> > >
> > > 1. **Expand factual-accuracy analysis beyond math to other MT-Bench domains.** Aside from math and reasoning, the other MT-Bench domains are either not reliably verifiable with automated methods (extraction, STEM, humanities, coding) or are not objectively verifiable at all (writing, roleplay). Accordingly, we had responses from all domains evaluated by expert human annotators, blind to the attack, and instructed to carefully assess the correctness and/or quality of each response. The results of this accuracy/quality audit were presented in our responses to Reviewers nFn5 and 7h5s, who explicitly requested it, and are reproduced below for your convenience. Due to the tight discussion period timeline, this analysis has so far been conducted by three expert annotators on a 25% sample of turn-1 responses from both our preamble pipeline and the non-attacked model. As agreed with the other reviewers, the study will be expanded to include all responses for the camera-ready version.
> > >
> > > 2. **Release cost estimation scripts or logs.** We agree, and will add an appendix detailing the calculations and exact final training cost of our pipeline.
> > >
> > > 3. **Discuss feasibility and trade-offs of training with an ensemble of judges.** Absolutely. As agreed in the rebuttal, we will discuss this in the paper as a current limitation and future work direction.
> > >
> > > 4. **Outline possible detection heuristics beyond perplexity filtering.** We are compiling references to support this discussion, which we outlined in the rebuttal, and we commit to adding it to Section 5.2 ‘Attack Detectability’.
> > >
> > > 5. **Strengthen societal-impact statement with a clear disclosure plan.** We will expand the discussion about attack exploitability in the Broader Impacts section with a note about model release/disclosure, as follows:
> > >
> > > - *“This work focuses on aligning candidate-LLMs to judge-LLMs by means of tuned preambles injected into the candidate, with the aim of obtaining inflated evaluations. While there is a chance that this strategy may be exploited by adversaries, it is of scientific interest to the community that such an attack is not only possible, but also particularly effective. Note that, while we openly disclose our training algorithm and hyperparameters and train using publicly available data, we do not release our trained preamble generator checkpoints to the public, as this may encourage their misuse.”*
> > >
> > > # MT-Bench accuracy/quality audit via expert human evaluation
> > >
> > > Firstly, each expert annotator assessed the correctness/quality of each response and labelled it as 'Poor', 'Fair' or 'Good'. We show the number of each label assigned to non-attacked and attacked responses in the table below. The distributions are fairly similar for both the non-attacked and the attacked model, with the non-attacked model receiving slightly more labels at either end of the scale ('Poor' and 'Good'), and the attacked one receiving slightly more mid-range labels ('Fair'). We observe relatively strong inter-annotator agreement (aggregated Spearman’s $\rho = 0.77$, combined $p < 0.001$, Kendall’s $W = 0.67$, $p < 0.001$).
> > >
> > > ||Poor|Fair|Good|
> > > |-|-|-|-|
> > > |No attack|20|6|34|
> > > |Preambles|19|9|32|
> > >
> > > Additionally, the annotators assigned a discrete 1-10 rating to each response (the same scale used by the judge-LLMs in MT-Bench). Below we show the average ratings obtained by each setup. The difference between the human ratings of the non-attacked and the attacked model is small (0.05), whereas the difference between the judge-LLM’s ratings of the same two setups is substantially greater in favour of our attack.
> > >
> > > ||Avg. human rating of responses|
> > > |-|-|
> > > |No attack|6.03|
> > > |Preambles|6.08|
> > >
> > > The above study will be expanded to include all responses for the camera-ready.
> > >
> > > We hope we have addressed the weaknesses you raised regarding the factual-accuracy audit, now extended to all domains via expert assessment, and the ethical-risk discussion, which we have expanded upon above.
> > >
> > > As for the attack detectability human study, which is currently relatively small-scale, we have reviewed the relevant literature and plan to expand this to 21 annotators and 525 samples in line with established prior work on LLM attacks [1, 2]. We will be able to do this in time for the camera-ready deadline in October. For the time being, however, we can only present the results already in the paper for 7 annotators and 175 samples, since this large-scale human study will take some time to set up and run.
> > >
> > > Thank you again for your positive engagement and constructive suggestions.
> > >
> > > Kind regards,\
> > > The Authors of Submission #23360
> > >
> > > ---
> > >
> > > [1] (Hu et al., 2024) Explaining Length Bias in LLM-Based Preference Evaluations.\
> > > [2] (Chu et al., 2025) JailbreakRadar: Comprehensive Assessment of Jailbreak Attacks Against LLMs.

---

### Official Review · Reviewer_eyDG · 2025-07-07

**Clarity:** 3
**Significance:** 3
**Originality:** 3
**Rating:** 5
**Confidence:** 3

**Summary:**

The paper proposes a method to generate preambles (equivalent to system prompts) which when attached as part of instructions to a frozen candidate LLM generates higher scored responses from a judge LLM. This is in contrast with previous approaches to game LLM judges by post hoc modification responses.

**Questions:**

Line 250: Missing reference to Table.

Appendix G.2 and G.3 (Preamble 6 and 9): Is there an explanation to why the Llama 8B+70B preambles do not look like natural English text as the other ones. I would assume the perplexity filter would easily flag these. It would be great to have some clarification.

**Ethical Concerns:**

["NO or VERY MINOR ethics concerns only"]

**Final Justification:**

The paper proposes a way to fool LLM judges which is creative. I would be fine if this paper is asked to be improved for a later venue to accept some other deserving paper.

**Limitations:**

yes

**Quality:**

3

**Strengths And Weaknesses:**

The paper has decent clarity of language and the experiment design and execution is adequate. There is originality in the idea of trying out a new way to fool LLM judges without post-hoc modification of candidate LLM responses. The baselines used are extensive and there is a discussion on transferability of proposed attacks.

---

> ### Author Rebuttal · Authors · 2025-07-30
>
> Dear Reviewer eyDG,
>
> We would like to thank you for taking the time to review our paper. We are glad that you found the experiment design and execution adequate and the idea original. Please find our responses to your comments and questions below.
>
> > **Question 1:** *“Line 250: Missing reference to Table.”*
>
> - Thank you for spotting this. It will be fixed immediately.
>
> > **Question 2:** *“Appendix G.2 and G.3 (Preamble 6 and 9): Is there an explanation to why the Llama 8B+70B preambles do not look like natural English text as the other ones. I would assume the perplexity filter would easily flag these. It would be great to have some clarification.”*
>
> - We believe the difference in fluency is mainly due to the setup in which we tune the KL-divergence weight $\beta$. We perform all hyperparameter tuning on the Command pipeline (due to computational limitations) and apply the same hyperparameters to the Llama pipeline. These include $\beta$, which regulates the faithfulness of the token distributions to those of the reference model, and thus the fluency of the preambles. Note that we prioritise downstream reward over fluency, since our preambles do not necessarily need to be human-readable, leading to a relatively low value of $\beta$ (as discussed in Appendix B.2 of the paper). We found that, using this low $\beta$ optimised on the Command pipeline, Llama's fluency steers away from the reference model faster and more decisively, while the downstream rewards increase. This behaviour may be due to differences in pre/post-training strategies, which tend to differ across distinct model families. Overall, we consider the Llama results particularly noteworthy, as they show that unfluent preambles tuned to produce high rewards are indeed effective in this context.
>
> - Indeed, unfluent preambles are not flagged by the perplexity filter. This is because the perplexity filter assesses the candidate LLM’s outputs (which are what the judge is able to see), and not the preambles which are part of its input (hidden from the judge). Analysing the LLM outputs is standard in prior literature and consistent with the intended use of the PPL filter detection method introduced by [1]. Remarkably, the frozen LLM’s outputs remain fluent even when we inject into it these unfluent preambles.
>
>
> We hope the responses above answer your questions. Please do let us know if you have other questions or comments.
>
> Kind regards,\
> The Authors of Submission #23360
>
> ---
> [1] (Jain et al., 2023) Baseline Defenses for Adversarial Attacks Against Aligned Language Models

---

> > ### Comment · Reviewer_eyDG · 2025-08-05
> >
> > Thanks for addressing my concerns.

---

> > > ### Author Response · Authors · 2025-08-05
> > >
> > > Dear Reviewer eyDG,
> > >
> > > Thank you for responding to our rebuttal. We are glad that our reply has addressed your concerns.
> > >
> > > Kind regards,\
> > > The Authors of Submission #23360

---

### Note · Authors · 2025-08-13

We would like to thank the reviewers for their thoughtful feedback and for recognising the value of our work. We are pleased that they found the idea *'novel'* and *'original'* (eyDG and 7h5s), the baseline comparison *'strong'* and *'extensive'* (eyDG and FfKE), the methodology *'sound'* (FfKE), the experiments and analysis *'comprehensive'* and *'thorough'* (FfKE and nFn5), and the presentation *'clear'* and *'precise'* (eyDG, FfKE and nFn5).

The feedback was highly constructive, and we believe the submission is now more well-rounded and robust, thanks to the additional experiments and clarifications prompted by the reviewers’ suggestions. Below, we summarise the main outcomes of the author-reviewer exchange. These new results and discussions are detailed in the individual rebuttals and will be integrated into the revised paper.

**Additional experiments and metrics**
- Evaluation on a new benchmark (*Arena-Hard*), demonstrating transferability without targeted training.
- Expanded transferability study with two additional LLM judges (in addition to the two in the original paper).
- Statistical analysis of results using paired-bootstrap confidence intervals.
- Factual accuracy scores for LLM answers in verifiable domains of the evaluation dataset.
- Expert human assessment scores for all LLM answers in both verifiable and non-verifiable domains (25% sample completed so far, extension to 100% in progress).

**Additional discussions**
- In-depth discussion of why the preambles in the Llama pipeline are substantially less fluent than the others.
- Feasibility and trade-offs of training with an ensemble of multiple LLM judges.
- Potential detection heuristics beyond perplexity filtering.
- Precise quantification of the API cost of our setup.
- Extended *Broader Impacts* section, now including a discussion on model release/disclosure.

**Other changes**
- Adjusted the plot colours for greyscale readability.
- Moved the most crucial hyperparameters from the Appendix to the main text.
- Added Appendix tables showing the fine-grained rewards per domain sub-category under each attack.
- Corrected three typos and further streamlined the language throughout.

Finally, as discussed with Reviewer FfKE, we plan to expand the human detectability study from 7 annotators and 175 responses to 21 annotators and 525 responses.

We thank again all reviewers for their valuable suggestions, active participation, and contributions to strengthening the quality of the work.

---

### Decision · Program_Chairs · 2025-09-17

**Decision:**

Accept (spotlight)

**Comment:**

My recommendation is to accept the paper as a spotlight.

The paper proposes a method for adversarially improving feedback in LLM-as-a-judge settings by training a model to provide a prompt-specific pre-amble to a frozen candidate LLM, whose output in response to the preamble + prompt is then sent to the judge. The authors show that this kind of attack has similar effectiveness to previous attacks that edit the candidate model's response after seeing the judge's feedback. The fact that the modifications happen to the input of the LLM make the attack harder to detect, a claim supported by a human study. The attack also transfers across candidate and judge LLMs.

Reviewers agreed that the paper showed creativity, precision in communication, and thoroughness in evaluation. There was consensus to accept after the rebuttal period, during which the authors were able to add additional support for their approach. I hope the authors will follow through on the improvements they promised during the rebuttal period, but even the paper as it was submitted had a combination of a fresh idea, thorough investigation, and solid execution that made it a good candidate for a spotlight.

I personally found the potential use for constructive goals to also be interesting.